# Unified Embedding: Battle-Tested Feature Representations for Web-Scale ML Systems

**Benjamin Coleman**[*]
Google DeepMind
colemanben@google.com

**Wang-Cheng Kang**[*]
Google DeepMind
wckang@google.com

**Matthew Fahrbach**
Google Research
fahrbach@google.com

**Ruoxi Wang**
Google DeepMind
ruoxi@google.com

**Lichan Hong**
Google DeepMind
lichan@google.com

**Ed H. Chi**
Google DeepMind
edchi@google.com

**Derek Zhiyuan Cheng**
Google DeepMind
zcheng@google.com

## Abstract

Learning high-quality feature embeddings efficiently and effectively is critical for the performance of web-scale machine learning systems. A typical model ingests hundreds of features with vocabularies on the order of millions to billions of tokens. The standard approach is to represent each feature value as a $d$-dimensional embedding, introducing hundreds of billions of parameters for extremely high-cardinality features. This bottleneck has led to substantial progress in alternative embedding algorithms. Many of these methods, however, make the assumption that each feature uses an independent embedding table. This work introduces a simple yet highly effective framework, *Feature Multiplexing*, where one single representation space is used across many different categorical features. Our theoretical and empirical analysis reveals that multiplexed embeddings can be decomposed into components from each constituent feature, allowing models to distinguish between features. We show that multiplexed representations lead to Pareto-optimal parameter-accuracy tradeoffs for three public benchmark datasets. Further, we propose a highly practical approach called *Unified Embedding* with three major benefits: simplified feature configuration, strong adaptation to dynamic data distributions, and compatibility with modern hardware. Unified embedding gives significant improvements in offline and online metrics compared to highly competitive baselines across five web-scale search, ads, and recommender systems, where it serves billions of users across the world in industry-leading products.

## 1   Introduction

There have been many new and exciting breakthroughs across academia and industry towards large-scale models recently. The advent of larger datasets coupled with increasingly powerful accelerators has enabled machine learning (ML) researchers and practitioners to significantly scale up models. This scale has yielded surprising emergent capabilities, such as human-like conversation and reasoning (Bubeck et al., 2023; Wei et al., 2022a). Interestingly, large-scale model architectures are headed towards two extreme scenarios. In natural language understanding, speech, and computer vision, the transformer (Vaswani et al., 2017) dominates the Pareto frontier. Most transformer parameters are

---

[*]Equal contribution.

37th Conference on Neural Information Processing Systems (NeurIPS 2023).

located in the hidden layers, with very small embedding tables (e.g., ~0.63B embedding parameters in the 175B GPT-3 (Brown et al., 2020)). On the other hand, search, ads, and recommendation (SAR) systems require staggering scale for state-of-the-art performance (e.g., 12T parameters in Mudigere et al. (2022)), but here, most of the parameters are in the embedding tables. Typical downstream hidden layers are 3 to 4 orders of magnitude smaller than the embedding tables. Recent work blurs the boundary between these extremes—large transformers started adopting bigger embedding tables for new types of tokens, and new SAR models leverage deeper and ever more-complicated networks. Therefore, embedding learning is a core technique that we only expect to become more critical and relevant to large-scale models in the future.

Motivated by web-scale ML for SAR systems, we study the problem of learning embeddings for categorical (sparse) features.[1] The weekend golf warriors in the ML community may appreciate our analogy between embeddings and the golf club grip (Hogan and Wind, 1985). In golf, the quality of your grip (feature embeddings) dictates the quality of your swing (model performance). Feature embeddings are the only point of contact between the data and the model, so it is critical to learn high-quality representations.

Feature embedding learning has been a highly active research area for nearly three decades (Rumelhart et al., 1986; Bengio et al., 2000; Mikolov et al., 2013; Papyan et al., 2020). In the linear model era, sparse features were represented as one-hot vectors. Linear sketches for dimension reduction became a popular way to avoid learning coefficients for each coordinate (Weinberger et al., 2009). In deep learning models, sparse features are not sketched but rather represented directly as embeddings. Learned feature embeddings carry a significant amount of semantic information, providing avenues to meaningful model interactions in the high-dimensional latent space.

**In practice**  The simplest transformation is to represent each feature value by a row in an $N \times d$ matrix (i.e., an *embedding table*). This approach is sufficient for many language tasks, but SAR features often have a massive vocabulary (e.g., tens of billions of product IDs in commerce) and highly dynamic power-law distributions. It is impractical to store and serve the full embedding table.

The *"hashing trick"* (Moody, 1988; Weinberger et al., 2009) provides a workaround by randomly assigning feature values to one of $M$ rows using a hash function. *Hash embeddings* (?) are a variation on the hashing trick where each feature value is assigned to multiple rows in the table. The final embedding is a weighted sum of rows, similar to a learned form of two-choice hashing (Mitzenmacher, 2001). *Compositional embeddings* (Shi et al., 2020) look up the feature in several independent tables and construct the final embedding from the components by concatenation, addition, or element-wise product. *HashedNet* embeddings (Chen et al., 2015) independently look up each dimension of the embedding in a flattened parameter space. *ROBE embeddings* (Desai et al., 2022) look up chunks, instead of single dimensions, to improve cache efficiency. Tsang and Ahle (2022) learn a sparse matrix to describe the row/dimension lookups, and *deep hash embeddings* (Kang et al., 2021) use a neural network to directly output the embedding.

**In theory**  The analysis of embedding algorithms is heavily influenced by linear models and the work of Weinberger et al. (2009) on the hashing trick. This line of reasoning starts with the assumption that all inputs have a "true" representation in a high-dimensional space that preserves all of the properties needed for learning (e.g., the one-hot encoding). The next step is to perform linear dimension reduction to compress the ground-truth representation into a smaller embedding, while approximately preserving inner products and norms. The final step is to extend the inner product argument to models that only interact with their inputs through inner products and norms (e.g., kernel methods).

In this framework, a key opportunity for improving performance is to reduce the dimension-reduction error. Early methods were based on the Johnson–Lindenstrauss lemma and sparse linear projections (Achlioptas, 2003; Li et al., 2006; Weinberger et al., 2009). Since then, a substantial effort has gone into improving the hash functions (Dahlgaard et al., 2017) and tightening the theoretical analysis of feature hashing (Dasgupta and Gupta, 2003; Freksen et al., 2018). Learned embeddings are now state-of-the-art, but the focus on dimension reduction remains: most recent algorithms are motivated by dimension reduction arguments. Previous attempts to understand hash embeddings do not study the full give-and-take relationship between feature encoding and the downstream learning problem. In this work, we carry the analysis forward to better understand learned embeddings.

---

[1]For simplicity, we call this problem "feature embedding learning," but we are primarily concerned with representing sparse features such as ZIP codes, word tokens, and product IDs.

**Contributions**   We present an embedding learning framework called *Feature Multiplexing*, where multiple features share one representation space. Our theoretical analysis and experimental studies on public benchmark datasets show that Feature Multiplexing is theoretically sound and generalizable to many popular embedding methods. We also propose a highly-efficient multiplexed approach called *Unified Embedding*, which is battle-tested and serves billions of users in industry-leading products via various web-scale search, ads, and recommendation models.

1. **Feature multiplexing**: We propose the Feature Multiplexing framework, where a single representation space (embedding table) is used to support different sparse features. All of the SOTA benchmark embedding algorithms in our experiments show better results when adopting Feature Multiplexing. Furthermore, the Unified Embedding approach (i.e., multiplexing + hashing trick) outperforms many SOTA embedding methods that are noticeably more complicated to implement and impractical for low-latency serving systems on modern day hardware.

2. **Analysis beyond collision counts**: We analyze hash embeddings in the standard framework as well as in a supervised learning setting to capture interactions between feature embedding and downstream learning. Our analysis gives insights and testable predictions about unified embeddings that cannot be obtained by dimension reduction arguments alone.

3. **Battle-tested approach and industrial experiments at scale**: Multi-size Unified Embedding has been launched in over a dozen web-scale SAR systems, significantly improving user engagement metrics and key business metrics. In addition, Unified Embedding presents major practical benefits: simplified feature configuration, strong adaptation to dynamic data distributions, and compatibility with ML accelerators. In Section 5.2, we share our insights, challenges, and practical lessons on adopting Unified Embedding.

## 2   Preliminaries

SAR applications often present supervised learning problems over categorical (sparse) features that represent properties of users, queries, and items. Each categorical feature consists of one or more *feature values* $v$ drawn from the *vocabulary* of the feature $V$. For a concrete example, consider the classic click-through rate (CTR) prediction task for ads systems, where we want to predict the probability of a user clicking on an advertisement. The "`ad_id`" feature may take on hundreds of billions of possible values—one for each unique ad served by the platform—while "`site_id`" describes the website where that ad was shown (a.k.a., the publisher).

We embed the values of each feature into a space that is more suitable for modeling, e.g., $\mathbb{R}^d$ or the unit $(d-1)$-sphere. Given a vocabulary of $N$ tokens $V = \{v_1, v_2, \ldots, v_N\}$, we learn a transformation that maps $v \in V$ to an embedding vector $\mathbf{e} \in \mathbb{R}^d$. After repeating this process for each categorical feature, we get a set of embeddings that are then concatenated and fed as input to a neural network (Covington et al., 2016; Anil et al., 2022).

**Problem statement**   We are given a dataset $D = \{(\boldsymbol{x}_1, y_1), (\boldsymbol{x}_2, y_2), \ldots, (\boldsymbol{x}_{|D|}, y_{|D|})\}$ of examples with labels. The examples consist of values from $T$ different categorical features with vocabularies $\{V_1, V_2, \ldots, V_T\}$, where feature $t$ has vocabulary $V_t$. Unless otherwise stated, we do not consider multivalent features and instead assume that each example $\boldsymbol{x}$ has one value for each categorical feature, i.e., $\boldsymbol{x} = [v_1, v_2, \ldots, v_T]$ where $v_i \in V_i$. However, this is for notational convenience only—the techniques in this paper extend to missing and multivalent feature values well.

Formally, the embedding table is a matrix $\mathbf{E} \in \mathbb{R}^{M \times d}$ and the embedding function $g(\mathbf{x}; \mathbf{E})$ transforms the categorical feature values into embeddings. We let $h(v) : V \to [M]$ be a 2-universal hash function (Vadhan et al., 2012) that assigns a feature value to a row index of the embedding table. We also consider a model function $f(\mathbf{e}; \boldsymbol{\theta})$ that transforms the concatenated embeddings into a model prediction, which is penalized by a loss $\ell$. Using this decomposition, we define the joint feature learning problem as follows:

$$\arg\min_{\mathbf{E}, \boldsymbol{\theta}} \mathcal{L}_D(\mathbf{E}, \boldsymbol{\theta}), \quad \text{where} \quad \mathcal{L}_D(\mathbf{E}, \boldsymbol{\theta}) = \sum_{(\boldsymbol{x}, y) \in D} \ell(f(g(\boldsymbol{x}; \mathbf{E}); \boldsymbol{\theta}), y). \tag{1}$$

We use independent hash functions $h_t(v)$ for each feature $t \in [T]$ so that tokens appearing in multiple vocabularies have the chance to get mapped to different locations, depending on the feature. We use the notation $\mathbf{e}_m$ to denote the $m$-th row of $\mathbf{E}$, which means $\mathbf{e}_{h(u)}$ is the embedding of $u$. Lastly, we let $\mathbb{1}_{u,v}$ be the indicator variable that denotes a hash collision between $u$ and $v$, i.e., $h(u) = h(v)$.

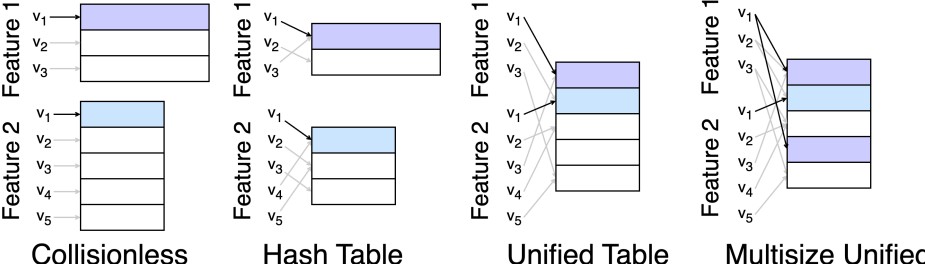

Figure 1: Embedding methods for two categorical features. We highlight the lookup process for the first value $v_1$ of each feature. Hash tables randomly share representations within each feature, while Unified Embedding shares representations across features. To implement Unified Embedding with different dimensions (multi-size or variable-length), we perform multiple lookups and concatenate the results.

## 3  Feature Multiplexing: The Inter-Feature Hashing Trick

It is standard practice to use a different hash table for each categorical feature. There are a few approaches that deviate from this common practice—for example, HashedNet and ROBE-Z models use parameter sharing across all features, depending on the implementation. However, conventional wisdom suggests that each categorical vocabulary benefits from having an independent representation, so shared representations are not well-studied.

**Parameter reuse**  We present a *Feature Multiplexing* framework that only uses one embedding table to represent all of the feature vocabularies. In contrast, typical models in SAR systems have hundreds of embedding tables, each representing one input feature. Multiplexed embeddings allow models to reuse parameters and lookup operations, improving efficiency in terms of both space and time. This is critical for models that must fit into limited GPU/TPU memory or satisfy latency and serving cost constraints. In principle, any feature embedding scheme (e.g., hashing trick, multihash, compositional, or ROBE-Z) can be used as the shared representation for feature multiplexing. In Section 5, we experiment with multiplexed versions of six different embedding algorithms. Here, we focus on a multi-size (multi-probe) version of the multiplexed feature hashing trick (Figure 1), which we call *Unified Embedding*. Unified Embedding presents unique practical advantages that we describe in Section 5.2.

**Tunable embedding width**  By using a unified table, we make the architectural assumption that all features share the same embedding dimension. However, this is often sub-optimal in practice since independent tuning of individual embedding dimensions can significantly improve model performance. For use-cases where different features require different embedding dimensions, we concatenate the results of a variable number of table lookups to obtain the final embedding. While this limits the output dimension to multiples of the table dimension, this is not a limiting constraint (and is needed by other methods such as ROBE-Z and product quantization).

**Shared vocabulary**  It is common for categorical feature vocabularies to share tokens. For example, two features that represent character bigrams of different text fields (e.g., query and document) might have substantial overlap. Our default approach is to use a different hash function for each feature by using a different hash seed for each feature. However, in cases where many semantically-similar features are present, the performance may slightly improve by using the same hash function.

## 4  Analysis of Feature Multiplexing

In this section, we present theoretical arguments about feature multiplexing applied to the "multiplexed feature hashing trick" setting. We start with a brief analysis in the dimension-reduction framework, suggesting that multiplexing can optimally load balance feature vocabularies across buckets in the table. This classic framework, however, cannot be used to distinguish between collisions that happen within the vocabulary of a feature (i.e., *intra-feature*) and those occurring across features (i.e., *inter-feature*).

To understand these interactions, we also need to consider the learning task and model training. Thus, we analyze the gradient updates applied to each bucket under a simplified but representative model—binary logistic regression with learnable feature embeddings for CTR prediction. Our analysis demonstrates a novel relationship between the embedding parameters and model parameters during gradient descent. Specifically, we find that the effect of inter-feature collisions can be mitigated if the model projects different features using orthogonal weight vectors. In Section 4.2, we empirically verify this by training models on public benchmark on click-through prediction (Criteo) and finding that the weight vectors do, in fact, orthogonalize. We defer the proofs and detailed derivations in this section to Appendix A.

## 4.1 Parameter Efficiency in The Dimension-Reduction Framework

In the feature hashing problem, we are given two vectors $\mathbf{x}, \mathbf{y} \in \mathbb{R}^N$ that we project using a linear map $\phi : \mathbb{R}^N \to \mathbb{R}^M$ given by the sparse $\{\pm 1\}$ matrix in Weinberger et al. (2009). The goal is for $\phi$ to minimally distort inner product, i.e., $\langle \phi(\mathbf{x}), \phi(\mathbf{y}) \rangle \approx \langle \mathbf{x}, \mathbf{y} \rangle$. Since the estimation error directly depends on the variance, it is sufficient to compare hashing schemes based on the moments of their inner product estimators. We examine the variance of the hashing trick and multiplexing trick.

To analyze multiple features at once in this framework, we consider the concatenations $\mathbf{x} = [\mathbf{x}_1, \mathbf{x}_2]$ and $\mathbf{y} = [\mathbf{y}_1, \mathbf{y}_2]$. The components $\mathbf{x}_1 \in \{0,1\}^{N_1}$ and $\mathbf{x}_2 \in \{0,1\}^{N_2}$ are the one-hot encodings (or bag-of-words) for the two vocabularies $V_1$ and $V_2$. The standard hashing trick approximates $\langle \mathbf{x}, \mathbf{y} \rangle$ by independently projecting $\mathbf{x}_1$ to dimension $M_1$ and $\mathbf{x}_2$ to dimension $M_2$. The multiplexed version uses a single projection, but into dimension $M_1 + M_2$. Now, we formalize the hashing trick of Weinberger et al. (2009), but with slightly revised notation. Given a set of tokens from the vocabulary, the embedding is the signed sum of value counts within each hash bucket.

**Definition 4.1.** Let $h : V \to \{1, 2, \ldots, M\}$ be a 2-universal hash function and $\xi : V \to \{-1, +1\}$ be a sign hash function. The function $\phi_{h,\xi} : 2^V \to \mathbb{R}^M$ is defined as $\phi_{h,\xi}(W) = \sum_{w \in W} \xi(w) \mathbf{u}_{h(w)}$, where $\mathbf{u}_i$ is the $i$-th unit basis vector in $\mathbb{R}^M$.

The following result compares multiplexed and standard embeddings in the dimension reduction framework. Note that the randomness is over the choice of hash functions $h(v)$ and $\xi(v)$. This result also directly extends to concatenating $T$ feature embeddings.

**Proposition 4.2.** *For any $\mathbf{x}_1, \mathbf{y}_1 \in \{0,1\}^{N_1}$ and $\mathbf{x}_2, \mathbf{y}_2 \in \{0,1\}^{N_2}$, let $\mathbf{x} = [\mathbf{x}_1, \mathbf{x}_2]$, $\mathbf{y} = [\mathbf{y}_1, \mathbf{y}_2]$ denote their concatenations. Let $\mu_U, \mu_H, \sigma_U^2$, and $\sigma_H^2$ be the mean and variance of $\langle \phi_{h,\xi}(\mathbf{x}), \phi_{h,\xi}(\mathbf{y}) \rangle$ for multiplexed and hash encodings, respectively. Then, $\mu_U = \mu_H = \langle \mathbf{x}, \mathbf{y} \rangle$ and*

$$\sigma_U^2 = \frac{\|\mathbf{x}\|_2^2 \|\mathbf{y}\|_2^2 + \langle \mathbf{x}, \mathbf{y} \rangle^2 - 2\langle \mathbf{x}, \mathbf{y} \rangle}{M_1 + M_2},$$

$$\sigma_H^2 = \frac{\|\mathbf{x}_1\|_2^2 \|\mathbf{y}_1\|_2^2 + \langle \mathbf{x}_1, \mathbf{y}_1 \rangle^2 - 2\langle \mathbf{x}_1, \mathbf{y}_1 \rangle}{M_1} + \frac{\|\mathbf{x}_2\|_2^2 \|\mathbf{y}_2\|_2^2 + \langle \mathbf{x}_2, \mathbf{y}_2 \rangle^2 - 2\langle \mathbf{x}_2, \mathbf{y}_2 \rangle}{M_2}.$$

**Observations** Proposition 4.2 shows that multiplexed embeddings can do a good job balancing hash collisions across the parameter space. Consider the problem of estimating the inner product $\langle \mathbf{x}, \mathbf{y} \rangle$. Suppose that $\|\mathbf{x}_1\|_2^2 = \|\mathbf{y}_1\|_2^2 = k_1$ and $\|\mathbf{x}_2\|_2^2 = \|\mathbf{y}_2\|_2^2 = k_2$, i.e., features 1 and 2 are multivalent with $k_1$ and $k_2$ values, and assume $\mathbf{x}$ and $\mathbf{y}$ are orthogonal, i.e., $\langle \mathbf{x}, \mathbf{y} \rangle = 0$. Proposition 4.2 tells us that $\sigma_U^2 = (k_1 + k_2)^2 / (M_1 + M_2)$ and $\sigma_H^2 = k_1^2 / M_1 + k_2^2 / M_2$. The difference $\sigma_H^2 - \sigma_U^2$ factors as $(\sqrt{M_1/M_2} k_1 - \sqrt{M_2/M_1} k_2)^2 / (M_1 + M_2)$, which is non-negative, implying $\sigma_U^2 \leq \sigma_H^2$.

## 4.2 Trajectories of Embeddings During SGD for Single-Layer Neural Networks

Intuition suggests that inter-feature collisions are less problematic than intra-feature collisions. For example, if $x \in V_1$ collides with $y \in V_2$ but no other values collide across vocabularies, then the embeddings from other tokens in $V_1$ and $V_2$ are free to rotate around the shared embedding for $x, y$ to preserve their optimal similarity relationships.

We analyze a logistic regression model with trainable embeddings for binary classification (i.e., a single-layer neural network with hashed one-hot encodings as input). This corresponds to Eq. (1) where $f(\mathbf{z}; \boldsymbol{\theta})$ is the sigmoid function $\sigma_{\boldsymbol{\theta}}(\mathbf{z}) = 1/(1 + \exp(-\langle \mathbf{z}, \boldsymbol{\theta} \rangle))$, $\ell$ is the binary cross-entropy loss, and $y \in \{0, 1\}$ (e.g., click or non-click). We concatenate the embeddings of each feature, which means the input to $f(\mathbf{z}; \boldsymbol{\theta})$ is $\mathbf{z} = g(\boldsymbol{x}; \mathbf{E}) = [\mathbf{e}_{h_1(\boldsymbol{x}_1)}, \mathbf{e}_{h_2(\boldsymbol{x}_2)}, \ldots, \mathbf{e}_{h_T(\boldsymbol{x}_T)}]$, where $\mathbf{e}_{h_t(\boldsymbol{x}_t)}$ is

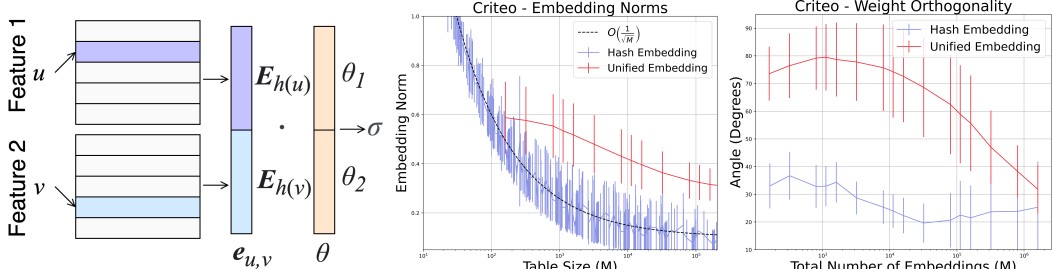

Figure 2: Single-layer neural embedding model with per-feature weights $\boldsymbol{\theta}_t$ (left). Mean embedding $\ell^2$-norm (middle) and mean angle between all pairs of weight vectors $\boldsymbol{\theta}_{t_1}, \boldsymbol{\theta}_{t_2}$ (right) as a function of table size for Criteo across all 26 categorical features. Note that the horizontal axes are in log scale.

the embedding for the $t$-th feature value in example $\boldsymbol{x}$. We write the logistic regression weights as $\boldsymbol{\theta} = [\boldsymbol{\theta}_1, \boldsymbol{\theta}_2, \ldots, \boldsymbol{\theta}_T]$ so that embedding $\mathbf{e}_{h_t(\boldsymbol{x}_t)}$ for feature $t$ is projected via $\boldsymbol{\theta}_t \in \mathbb{R}^M$. We illustrate this simple but representative model architecture in Figure 2.

The following analysis holds in general for $T$ features, but we consider the problem for two categorical features with vocabularies $V_1$ and $V_2$ to simplify the presentation. We write the logistic regression objective as follows in Eq. (2), using $C_{u,v,1}$ and $C_{u,v,0}$ to denote the number of examples for which $\boldsymbol{x} = [u, v]$ and $y = 1$ and $y = 0$, respectively. We present a full derivation in Appendix A.2.

$$\mathcal{L}_D(\mathbf{E}, \boldsymbol{\theta}) = - \sum_{u \in V_1} \sum_{v \in V_2} C_{u,v,0} \log \exp(\boldsymbol{\theta}^\top \mathbf{e}_{u,v}) - (C_{u,v,0} + C_{u,v,1}) \log(1 + \exp(\boldsymbol{\theta}^\top \mathbf{e}_{u,v})) \quad (2)$$

To understand the effect of hash collisions on training dynamics, we take the gradient with respect to $\mathbf{e}_{h_1(u)}$, i.e., the embedding parameters that represent $u \in V_1$. By studying the gradients for collisionless, hash, and multiplexed embeddings, we quantify the effects of intra-feature and inter-feature collisions. Interestingly, all three gradients can be written using the following three terms:

$$\nabla_{\mathbf{e}_{h_1(u)}} \mathcal{L}_D(\mathbf{E}, \boldsymbol{\theta}) = \boldsymbol{\theta}_1 \sum_{v \in V_2} C_{u,v,0} - (C_{u,v,0} + C_{u,v,1}) \sigma_{\boldsymbol{\theta}}(\mathbf{e}_{u,v}) \quad (3)$$

$$+ \boldsymbol{\theta}_1 \sum_{\substack{w \in V_1 \\ w \neq u}} \mathbb{1}_{u,w} \sum_{v \in V_2} C_{w,v,0} - (C_{w,v,0} + C_{w,v,1}) \sigma_{\boldsymbol{\theta}}(\mathbf{e}_{u,v}) \quad (4)$$

$$+ \boldsymbol{\theta}_2 \sum_{v \in V_2} \mathbb{1}_{u,v} \sum_{w \in V_1} C_{w,v,0} - (C_{w,v,0} + C_{w,v,1}) \sigma_{\boldsymbol{\theta}}(\mathbf{e}_{w,v}). \quad (5)$$

For collisionless embeddings, component (3) is the full gradient. For hash embeddings, the gradient is the sum of components (3) and (4). The multiplexed embedding gradient is the sum of all three terms.

**Insights** Our core observation is that embedding gradients can be decomposed into a collisionless (true) component (3), an intra-feature component (4), and an inter-feature component (5). The components from hash collisions act as a gradient bias. The intra-feature component is in the same direction as the true component, implying that intra-feature collisions are not resolvable by the model. This agrees with intuition since intra-feature hash collisions effectively act as value merging.

Inter-feature collisions, however, push the gradient in the direction of $\boldsymbol{\theta}_2$, which creates an opportunity for model training to mitigate their effect. To see this, consider a situation where $\boldsymbol{\theta}_1$ and $\boldsymbol{\theta}_2$ are initialized to be orthogonal and do not change direction during training.[2] During SGD, the embedding $\mathbf{e}_{h_1(u)}$ is a linear combination of gradients over the training steps, which means $\mathbf{e}_{h_1(u)}$ can be decomposed into a true and intra-feature component, in the $\boldsymbol{\theta}_1$ direction, and an inter-feature component in the $\boldsymbol{\theta}_2$ direction. Since $\langle \boldsymbol{\theta}_1, \boldsymbol{\theta}_2 \rangle = 0$, the projection of $\boldsymbol{\theta}_1^\top \mathbf{e}_{h_1(u)}$ effectively *eliminates the inter-feature component*.

---

[2]This restriction does not affect the representation capacity because learned embeddings can rotate around $\boldsymbol{\theta}$, i.e., for any $\boldsymbol{\theta}, \mathbf{e}$, and nonzero constant $\mathbf{u}$, we can learn some $\mathbf{e}'$ that satisfies $\langle \mathbf{e}', \mathbf{u} \rangle = \langle \mathbf{e}, \boldsymbol{\theta} \rangle$.

**Empirical study**    These observations allow us to form two testable hypotheses. First, we expect the projection vectors $[\boldsymbol{\theta}_1, \boldsymbol{\theta}_2, \ldots, \boldsymbol{\theta}_T]$ to orthogonalize since this minimizes the effect of inter-feature collisions (and improves the loss). The orthogonalization effect should be stronger for embedding tables with fewer hash buckets since that is when inter-feature collisions are most prevalent. Second, intra-feature and inter-feature gradient contributions add approximately $O(N/M)$ for each of their dimensions to the collisionless gradient since $\mathbb{E}[\mathbb{1}_{u,v}] = 1/M$. Therefore, we expect the squared embedding norms to scale roughly as $O(N/M)$.

To investigate these hypotheses, we train a single-layer neural network on the categorical features of the Criteo click-through prediction task. We initialize $\boldsymbol{\theta} = [\boldsymbol{\theta}_1, \boldsymbol{\theta}_2, \ldots, \boldsymbol{\theta}_T]$ with each $\boldsymbol{\theta}_t$ component in the same direction (i.e., the worst-case) to see if multiplexing encourages the weights to orthogonalize. We plot the results in Figure 2, which support our hypotheses.

### 4.3   Why Can Features Share a Single Table?

Returning to our parameter efficiency argument of Section 4.1, we can finally explain the full benefits of feature multiplexing. The dimension reduction argument shows that, given a parameter budget, the multiplexed embedding has the same overall number of hash collisions as a well-tuned hash embedding. Our gradient analysis, however, shows that not all collisions are equally problematic. Inter-feature collisions can be mitigated by a single-layer neural network because different features are processed by different model parameters. While our theoretical results are limited to single-layer neural networks, we expect deeper and more complicated network architectures to exhibit analogs of weight orthogonalization due to their relative overparameterization. However, we believe this is still compelling evidence that shared embedding tables work by load balancing unrecoverable collisions across a larger parameter space and effectively use one massive table for each feature.

## 5   Experiments

Finally, we present experimental results on three public benchmark datasets, followed by anonymized, aggregate results from our industrial deployments across multiple web-scale systems. We give more details in Appendix B, including full parameter-accuracy tradeoffs, additional experiments, and the neural network architectures used.

### 5.1   Experiments with Public Benchmark Datasets

**Datasets**    Criteo is an online advertisement dataset with ~45 million examples (7 days of data). Each example corresponds to a user interaction (click/no-click decision) and has 39 corresponding features. 26 of these features are categorical, and 13 are real-valued continuous variables. We only embed the categorical features. Avazu is a similar advertisement click dataset but with ~36 million examples (11 days of data). Each example here contains 23 categorical features and no continuous features. Movielens is a traditional user-item collaborative filtering problem, which we convert to a binary prediction task by assigning "1"s to all examples with rating $\geq 3$, and "0"s to the rest. For more details, see the benchmarks in Zhu et al. (2022) for Criteo and Avazu and the review by Harper and Konstan (2015) for Movielens. For Criteo and Movielens, we apply the same continuous feature transformations, train-test split, and other preprocessing steps in Wang et al. (2021). For Avazu, we use the train-test split and preprocessing steps in Song et al. (2019).

**Experiment design**    To find the Pareto frontier of the memory-AUC curve (Figure 3), we train a neural network with different embedding representation methods at 16 logarithmically-spaced memory budgets. Models can overfit on the second epoch, but compressed embeddings sometimes require multiple epochs. To showcase the best performance, we follow the guidelines of Tsang and Ahle (2022) and report the test performance of the best model found over three epochs. As baselines, we implement collisionless embeddings, the feature hashing trick, hash embeddings, HashedNets, ROBE-Z, compositional embeddings with element-wise product (QR), and compositional embeddings with concatenation (i.e., product quantization (PQ)).

For the multiplexed version of an existing embedding scheme, we use a single instance of the scheme to index all of the features. For example, we look up all feature values in a single multihash table, rather than using a separately-tuned multihash table for each feature. This is equivalent to salting the vocabulary of each feature with the feature ID and merging the salted vocabularies into a single massive vocabulary (which is then embedded as usual). We apply this procedure to each baseline to obtain a new multiplexed algorithm, reported below the horizontal bar in Table 1.

**Implementation and hyperparameters**   All methods are implemented in Tensorflow 2.0 using the TensorFlow Recommenders (TFRS) framework.[3] For a fair comparison, all models are identical except for the embedding component. Following other studies (Naumov et al., 2019; Wang et al., 2021), we use the same embedding dimension for all features ($d \in \{39, 32, 30\}$ for Criteo, Avazu, and Movielens, respectively). This is followed by a stack of 1-2 DCN layers and 1-2 DNN feed-forward layers—we provide a full description of the architecture in Appendix B. We run a grid search to tune all embedding algorithm hyperparameters and conduct five runs per combination. When allocating embedding parameters to features, we size tables based on the cardinality of their vocabulary (e.g., a feature with 10% of the total vocabulary receives 10% of the memory budget).

**Results**   Table 1 shows the results for the academic datasets. We give three points on the parameter-accuracy tradeoff and highlight the Pareto-optimal choices, but present the full tradeoff (including standard deviations across runs) in Appendix B. Figure 3 shows the Pareto frontier of the parameter-accuracy tradeoff for all multiplexed and non-multiplexed methods. We observe that feature multiplexing is Pareto-optimal. Interestingly, the multiplexed hashing trick outperforms several embedding techniques (e.g., ROBE-Z) that were previously SOTA.

Collisionless embeddings are included in our benchmark evaluation only to provide an optimistic headroom reference point. Collisionless tables are often impossible to deploy in industrial recommendation systems since feature vocabularies can be massive and change dynamically due to churn. Multiplexing provides a significant improvement to methods that are fast, implementable, and orders of magnitude cheaper to train and serve than collisionless embeddings.

For significance tests comparing multiplexed methods to their corresponding non-multiplexed baseline (rather than the strongest baseline), see Appendix B. Appendix B also contains additional details about Criteo and Avazu, including an explanation for the collisionless-hash embedding performance gap on Criteo.

Table 1: AUC of embedding methods on click-through datasets. We give results for several points on the parameter-accuracy tradeoff, listed as sub-headings under the dataset that describe the total embedding table size. Our multiplexed methods (listed below the bar) outperform the strongest baseline at $0.01^{**}$ and $0.05^*$ significance levels of Welch's $t$-test.

| Method | Criteo | | | Avazu | | | Movielens-1M | | |
|---|---|---|---|---|---|---|---|---|---|
| | 25MB | 12.5MB | 2.5MB | 32.4MB | 3.24MB | 324kB | 1.6MB | 791kB | 158kB |
| Collisionless | 80.70 | – | – | 77.35 | – | – | 88.72 | – | – |
| Hashing Trick | 80.21 | 79.98 | 79.44 | 77.24 | 76.71 | 75.10 | 85.37 | 83.00 | 77.07 |
| Hash Embedding | 79.82 | 80.34 | 80.40 | 77.29 | 76.89 | 76.86 | 88.28 | 87.71 | 86.05 |
| HashedNet | 80.54 | 80.50 | 80.42 | 77.30 | 77.19 | 76.89 | 87.92 | 87.92 | 87.31 |
| ROBE-Z Embedding | 80.55 | 80.52 | 80.41 | 77.32 | 77.18 | 76.88 | 88.05 | 87.92 | 87.40 |
| PQ Embedding | 80.29 | 80.25 | 79.82 | **77.39** | 77.11 | 76.48 | 88.49 | 88.38 | 87.32 |
| QR Embedding | 80.29 | 80.21 | 79.74 | 76.98 | 76.61 | 75.86 | 88.49 | 88.63 | 88.22 |
| Multiplex Hash Trick | 80.43 | 80.47 | 80.49$^{**}$ | 77.35 | 77.18 | 76.86 | 87.74 | 86.93 | 82.00 |
| Multiplex HashEmb | 80.52 | 80.42 | 80.31 | 77.28 | 77.16 | 76.82 | 88.79$^{**}$ | **88.74**$^*$ | 87.66 |
| Multiplex HashedNet | 80.53 | 80.55 | 80.48$^{**}$ | 77.30 | 77.23$^*$ | 76.94$^{**}$ | 87.95 | 87.93 | 87.57 |
| Multiplex ROBE-Z | **80.57** | **80.56** | 80.50$^*$ | 77.32 | 77.24$^{**}$ | 76.94$^{**}$ | 88.00 | 88.03 | 87.62 |
| Multiplex PQ | 80.43 | 80.49 | **80.54**$^{**}$ | **77.39** | **77.29**$^{**}$ | **76.96**$^{**}$ | 88.81$^{**}$ | 88.69$^*$ | 88.09 |
| Multiplex QR | 80.22 | 80.28 | 80.48$^{**}$ | 77.29 | 77.16 | 76.90 | **88.83**$^{**}$ | 88.53 | **88.27**$^{**}$ |

## 5.2   Industrial Deployments of Unified Embedding in Web-Scale Systems

Among the proposed approaches, Unified Embedding (multi-probe hashing trick + multiplexing) is uniquely positioned to address the infrastructure and latency constraints associated with training and serving web-scale systems. We adopt Unified Embedding in practice, using variable row lookups for each feature (typically 1~6) with concatenation to achieve the desired embedding width for our features. It should be noted that this process yields a similar algorithm to Multiplex PQ—one of the top performers from Table 1. We discuss how to adopt Unified Embedding in production SAR systems, where it has already served billions of users through launches in over a dozen models. We also present an experimental study of Unified Embedding in offline and online A/B experiments.

---

[3] https://github.com/tensorflow/recommenders

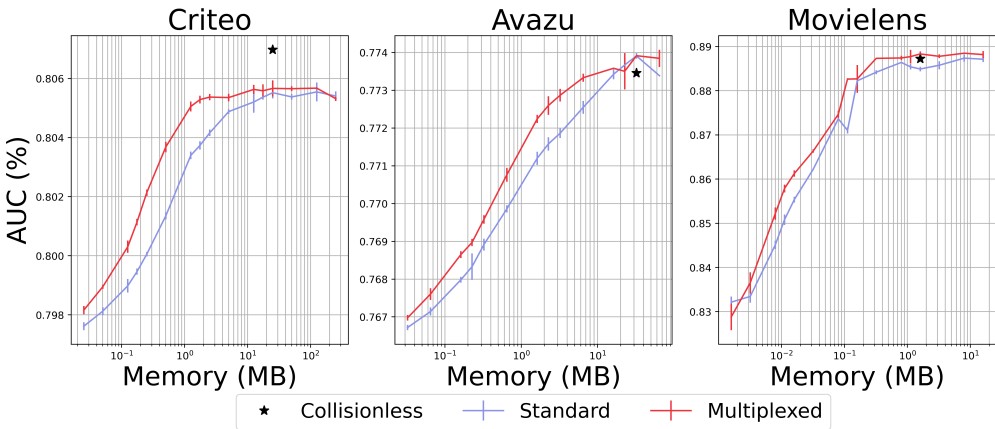

Figure 3: Pareto frontier of multiplexed and original (non-multiplexed) methods. Top-left is better.

**Experiment and production setup**   To test the robustness and generalization capability of Unified Embedding, we conduct offline and online experiments with five key models from different SAR domains, including commerce, apps, and short-form videos. These systems represent a diverse set of production model architectures powered by state-of-the-art techniques, including two-tower retrieval models with LogQ correction (Yi et al., 2019), DCN-V2 (Wang et al., 2021), and MMOE (Zhao et al., 2019). These models handle a wide range of prediction tasks, including candidate retrieval, click-through rate (pCTR), conversion rate (pCVR) prediction (Chapelle, 2014), and multi-task user engagement modeling.

All of these production models are continuously trained and updated in an online streaming fashion, with the embedding tables being the dominant component (often >99%) in terms of model parameters (Anil et al., 2022; Fahrbach et al., 2023). Many of the baseline embedding methods are simple but quite strong (e.g., feature hashing and multiple hashing), and have been tuned with AutoML and human heuristics (Anil et al., 2022). In fact, these baseline embedding learning methods have roughly stayed the same for the past five years, with various efforts failing to improve them through novel techniques.

**Practical benefits**   Unified Embedding tables offer solutions to several challenging and practical issues in large-scale embedding learning systems.

- **Simple configuration**: With standard embedding methods, we need to independently tune the table size and dimension for each feature. Hyperparameter tuning is an arduous process due to the large number of categorical features used in production (hundreds if not thousands). Unified Embedding tables are far simpler to configure: the total table size is calculated based on the available memory budget, and we only need to tune the dimension on a per-feature basis—easily accomplished via AutoML (Bender et al., 2020) or attention-based search methods (Yasuda et al., 2023; Axiotis and Yasuda, 2023). This yields roughly a 50% reduction in the number of hyperparameters.

- **Adapt to dynamic feature distributions**: In practice, the vocabulary size of each feature changes over time (new IDs enter the system on a daily basis, while stale items gradually vanish). For example, in a short-form video platform, we expect a highly dynamic video ID vocabulary with severe churn. In e-commerce, a significant number of new products are added during the holiday season compared to off-season. Thus, a fixed number of buckets for a given table can often lead to sub-optimal performance. Unified Embedding tables offer a large parameter space that is shared across all features, which can better accommodate fluctuations in feature distributions.

- **Hardware compatibility**: Machine learning accelerators and frameworks have co-evolved with algorithms to support common use-cases, and the "row embedding lookup" approach has been the standard technique for the past decade. While Unified Embedding tables are well-supported by the latest TPUs (Jouppi et al., 2023) and GPUs (Wei et al., 2022b; Mudigere et al., 2022), many of the recent and novel embedding methods require memory access patterns that are not as compatible with ML accelerators.

**Offline and online results**   Table 2 displays the results of our offline and online experiments for Unified Embedding. It was applied to production models with vocabulary sizes spanning from 10M to 10B. We observe substantial improvements across all models. Note that the offline metrics and the key business metrics for different models can be different, and the gains are not directly comparable. For offline metrics, the models are evaluated using AUC (higher is better), Recall@1 (higher is better), and RMSE (lower is better). From these experiments, we found that feature multiplexing provides greater benefits when (i) the vocabulary sizes for features are larger, and (ii) the vocabularies are more dynamic (e.g., suffers from a high churn rate). For instance, an overwhelming number of new short-form videos are produced daily, creating a large and dynamic vocabulary.

Table 2: Unified Embedding applied to different production model architectures and prediction tasks across various domains. Note that +0.1% is considered significant due to the large amount of traffic served in online A/B experiments. All multiplexed models are neutral or better in terms of model size and training/serving costs.

| Task | pCTR | Retrieval | Multi-Task Ranking | pCVR | pCTR |
|---|---|---|---|---|---|
| Domain | Products | Products | Short-form videos | Apps | Apps |
| Architecture | DCN-V2 | Two-tower | MMOE | DCN-V2 | DCN-V2 |
| Vocabulary Size | ~10B | ~10B | ~1B | ~10M | ~10M |
| Offline Metric | AUC +2.2% | Recall@1 +7.3% | AUC +0.39% (task 1) RMSE -0.53% (task 2) | AUC +0.25% | AUC +0.17% |
| Online Metric | Positive | Positive | +0.62% | +0.44% | +0.11% |

# 6   Conclusion

In this work, we propose a highly generalizable *Feature Multiplexing* framework that allows many different features to share the same embedding space. One of our crucial observations is that models can learn to mitigate the effects of inter-feature collisions, enabling a single embedding vector to represent feature values from different vocabularies. Multiplexed versions of SOTA embedding methods provide a Pareto-optimal tradeoff, and *Unified Embedding* proves to be a particularly simple yet effective approach. We demonstrate that a Unified Embedding table performs significantly better in both offline and online experiments compared to highly-competitive baselines across multiple web-scale ML systems. We believe the interplay between feature multiplexing, dynamic vocabularies, power-law feature distributions, and DNNs provides interesting and important opportunities for future work.

**Limitations**   While we expect deeper and more complicated network architectures to exhibit similar behavior, our theoretical analysis is limited to single-layer neural networks. The Pareto frontier from Section 5.1 is based on models that lag behind the current SOTA (we use 1-2 DCN layers + 1-2 DNN layers), though we provide evidence that feature multiplexing is also effective for SOTA models in Table 2. Unified embeddings impose limitations on the overall model architecture, and all embedding dimensions must share a common multiple. Because of the latency involved with looking up more than 6 components, we are often limited to 5-6 discrete choices of embedding width. Finally, unified embeddings sacrifice the ability to aggressively tune hyperparameters on a per-feature basis in exchange for flexibility, ease-of-use, and simplified feature engineering.

# Acknowledgements

The authors would like to thank our amazing collaborators in helping develop the research, integrate our approaches into production, and reviewing our paper. Here is the list of contributors in alphabetical order by last name: Shuchao Bi, Bo Chang, Kevin Chang, Shuo Chen, Chen Cheng, Chris Collins, Evan Ettinger, Minjie Fan, Gang Fu, Xing Gao, Eu-jin Goh, Matthew Gonzalgo, Cristos Goodrow, Wenshuo Guo, Zhongze Hu, Gui Huan, Da Huang, Yanhe Huang, Samuel Ieong, Arpith Jacob, Maciej Kula, George Kurian, Duo Li, Yang Li, Yueling Li, Silva Lin, Liang Liu, Wenjing Ma, Lifeng Nai, Abdul Saleh, Mahesh Sathiamoorthy, Edward Shen, Rakesh Shivanna, Jaideep Singh, Ryan Smith, Myles Sussman, Jiaxi Tang, Alice Xu, Luning Yang, Tiansheng Yao, Wenxing Ye, Xinyang Yi, Angel Yu, David Zats, Fan Zhang, Jerry Zhang, Tianyin Zhou, Yun Zhou, Qinyun Zhu.

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

# A Theory

## A.1 Missing Analysis from Section 4.1

**Lemma A.1** (Weinberger et al. (2009, Lemma 2)). *For any* $\mathbf{x}, \mathbf{y} \in \mathbb{R}^n$, $\mathbb{E}_{h,\xi}[\langle \phi(\mathbf{x}), \phi(\mathbf{y}) \rangle] = \langle \mathbf{x}, \mathbf{y} \rangle$. *Furthermore, we have*

$$\text{Var}_{h,\xi}(\langle \phi(\mathbf{x}), \phi(\mathbf{y}) \rangle) = \frac{1}{m} \sum_{i \neq j} x_i^2 y_j^2 + x_i y_i x_j y_j$$

$$= \frac{1}{m}(\langle \mathbf{x}, \mathbf{x} \rangle \langle \mathbf{y}, \mathbf{y} \rangle + \langle \mathbf{x}, \mathbf{y} \rangle \langle \mathbf{x}, \mathbf{y} \rangle - 2\langle \mathbf{x} \circ \mathbf{y}, \mathbf{x} \circ \mathbf{y} \rangle),$$

*where* $\mathbf{x} \circ \mathbf{y} \in \mathbb{R}^n$ *denotes the Hadamard product of* $\mathbf{x}$ *and* $\mathbf{y}$.

**Proposition 4.2.** *For any* $\mathbf{x}_1, \mathbf{y}_1 \in \{0, 1\}^{N_1}$ *and* $\mathbf{x}_2, \mathbf{y}_2 \in \{0, 1\}^{N_2}$, *let* $\mathbf{x} = [\mathbf{x}_1, \mathbf{x}_2]$, $\mathbf{y} = [\mathbf{y}_1, \mathbf{y}_2]$ *denote their concatenations. Let* $\mu_U$, $\mu_H$, $\sigma_U^2$, *and* $\sigma_H^2$ *be the mean and variance of* $\langle \phi_{h,\xi}(\mathbf{x}), \phi_{h,\xi}(\mathbf{y}) \rangle$ *for multiplexed and hash encodings, respectively. Then,* $\mu_U = \mu_H = \langle \mathbf{x}, \mathbf{y} \rangle$ *and*

$$\sigma_U^2 = \frac{\|\mathbf{x}\|_2^2 \|\mathbf{y}\|_2^2 + \langle \mathbf{x}, \mathbf{y} \rangle^2 - 2\langle \mathbf{x}, \mathbf{y} \rangle}{M_1 + M_2},$$

$$\sigma_H^2 = \frac{\|\mathbf{x}_1\|_2^2 \|\mathbf{y}_1\|_2^2 + \langle \mathbf{x}_1, \mathbf{y}_1 \rangle^2 - 2\langle \mathbf{x}_1, \mathbf{y}_1 \rangle}{M_1} + \frac{\|\mathbf{x}_2\|_2^2 \|\mathbf{y}_2\|_2^2 + \langle \mathbf{x}_2, \mathbf{y}_2 \rangle^2 - 2\langle \mathbf{x}_2, \mathbf{y}_2 \rangle}{M_2}.$$

*Proof.* The results for $\mu_U$, $\mu_H$ and $\sigma_U^2$ directly follow from Lemma A.1, and observing that

$$\langle \mathbf{x} \circ \mathbf{y}, \mathbf{x} \circ \mathbf{y} \rangle = \langle \mathbf{x} \circ \mathbf{x}, \mathbf{y} \circ \mathbf{y} \rangle = \langle \mathbf{x}, \mathbf{y} \rangle,$$

since $\mathbf{x}, \mathbf{y} \in \{0, 1\}^{N_1 + N_2}$.

To compute $\sigma_H^2$, first decompose the inner product

$$\langle \phi(\mathbf{x}), \phi(\mathbf{y}) \rangle = \langle \phi_{h_1,\xi_1}(\mathbf{x}_1), \phi_{h_1,\xi_1}(\mathbf{y}_1) \rangle + \langle \phi_{h_2,\xi_2}(\mathbf{x}_2), \phi_{h_2,\xi_2}(\mathbf{y}_2) \rangle.$$

Since the hash functions $h_1, \xi_1, h_2, \xi_2$ are independent, it follows that $\sigma_H^2$ is the sum of the variances of $\langle \phi_{h_1,\xi_1}(\mathbf{x}_1), \phi_{h_1,\xi_1}(\mathbf{y}_1) \rangle$ and $\langle \phi_{h_2,\xi_2}(\mathbf{x}_2), \phi_{h_2,\xi_2}(\mathbf{y}_2) \rangle$, so we can again invoke Lemma A.1. $\square$

## A.2 Derivation of Gradients

For easy reference, we re-introduce our logistic regression setting here. In Figure 4, we reproduce part of Figure 2 from the main text to further help clarify our notation.

Recall from the main text that we are interested in a logistic regression model with trainable embeddings for binary classification (i.e., a single-layer neural network with hashed one-hot encodings as input). This corresponds to our supervised learning framework (Eq. (1)), where $f(\mathbf{z}; \boldsymbol{\theta})$ is the sigmoid function $\sigma_{\boldsymbol{\theta}}(\mathbf{z}) = 1/(1 + \exp(-\langle \mathbf{z}, \boldsymbol{\theta} \rangle))$ and $\ell$ is the binary cross-entropy loss.

We concatenate the embeddings of each feature, which means that the input to $f(\mathbf{z}; \boldsymbol{\theta})$ is $\mathbf{z} = g(\mathbf{x}; \mathbf{E}) = [\mathbf{e}_{h_1(\mathbf{x}_1)}, \mathbf{e}_{h_2(\mathbf{x}_2)}, \ldots, \mathbf{e}_{h_T(\mathbf{x}_T)}]$, where $\mathbf{e}_{h_t(\mathbf{x}_t)}$ is the embedding for the $t$-th feature value in example $\mathbf{x}$.

We also partition the logistic regression weights $\boldsymbol{\theta} = [\boldsymbol{\theta}_1, \boldsymbol{\theta}_2, \ldots, \boldsymbol{\theta}_T]$, so that embedding $\mathbf{e}_{h_t(\mathbf{x}_t)}$ for feature $t$ is projected via $\boldsymbol{\theta}_t \in \mathbb{R}^M$.

For notational convenience we also partition the dataset by label. We write $D_0 = \{(\mathbf{x}_i, y_i) \in D : y_i = 0\}$ and $D_1 = \{(\mathbf{x}_i, y_i) \in D : y_i = 1\}$. Using this notation, we can write the logistic regression objective as follows:

$$\mathcal{L}_D(\mathbf{E}, \boldsymbol{\theta}) = - \sum_{(\mathbf{x},y) \in D_0} \log\left(\frac{1}{1 + \exp(-\langle \boldsymbol{\theta}, g(\mathbf{x}; \mathbf{E}) \rangle)}\right) - \sum_{(\mathbf{x},y) \in D_1} \log\left(\frac{1}{1 + \exp(\langle \boldsymbol{\theta}, g(\mathbf{x}; \mathbf{E}) \rangle)}\right).$$

We proceed by writing the loss as a sum over vocabulary pairs rather than examples. We consider the problem for two categorical features with vocabularies $V_1$ and $V_2$ to simplify the presentation, and we use the notation $C_{u,v,0}$ to denote the number of co-occurences of $(u, v) \in V_1 \times V_2$ in dataset $D_0$.

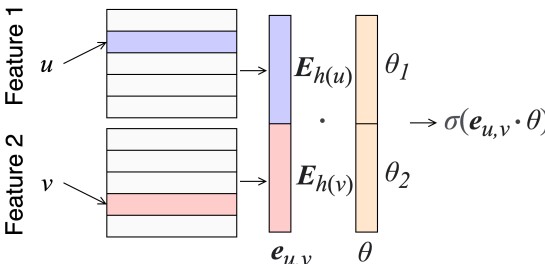

Figure 4: Illustration of the logistic regression model, annotated with notation that corresponds to our theoretical analysis.

Likewise, $C_{u,v,1}$ denotes the corresponding quantity for $D_1$. We also let $\mathbf{e}_{u,v} = [\mathbf{e}_{h_1(u)}, \mathbf{e}_{h_2(v)}]$ so that the inner product $\langle [\boldsymbol{\theta}_1, \boldsymbol{\theta}_2], [\mathbf{e}_{h_1(u)}, \mathbf{e}_{h_2(v)}] \rangle$ can be written concisely as $\boldsymbol{\theta}^\top \mathbf{e}_{u,v}$. It follows that

$$\mathcal{L}_D(\mathbf{E}, \boldsymbol{\theta}) = -\sum_{u \in V_1} \sum_{v \in V_2} C_{u,v,0} \log\left(\frac{1}{1 + \exp(-\boldsymbol{\theta}^\top \mathbf{e}_{u,v})}\right) + C_{u,v,1} \log\left(\frac{1}{1 + \exp(\boldsymbol{\theta}^\top \mathbf{e}_{u,v})}\right).$$

After combining the sigmoid functions, we get the following expression:

$$\mathcal{L}_D(\mathbf{E}, \boldsymbol{\theta}) = -\sum_{u \in V_1} \sum_{v \in V_2} C_{u,v,0} \log \exp(\boldsymbol{\theta}^\top \mathbf{e}_{u,v}) - (C_{u,v,0} + C_{u,v,1}) \log(1 + \exp(\boldsymbol{\theta}^\top \mathbf{e}_{u,v})).$$

To get the gradients with respect to $\mathbf{E}_{h(u)}$ (the embedding parameters used to represent value $u$), we first take the gradient with respect to $\mathbf{E}_m$. These parameters accumulate gradient contributions from any value assigned to bucket $m$ (i.e. for which $\mathbb{1}_{u,m}$ is one). It is then straightforward to express $\nabla_{\mathbf{E}_{h(u)}} \mathcal{L}_D(\mathbf{E}, \theta)$ in terms of the other values $v$ that collide with $u$ (i.e. for which $\mathbb{1}_{u,v}$ is one).

**Collisionless Embeddings:** Because each row $\mathbf{E}_m$ corresponds to a single value, we can take the gradient with respect to $\mathbf{E}_{h(u)}$ directly.

$$\nabla_{\mathbf{E}_{h(u)}} \mathcal{L}_D(\mathbf{E}, \theta) = \theta_1 \sum_{v \in V_2} C_{u,v,0} - (C_{u,v,0} + C_{u,v,1}) \sigma_\theta(\mathbf{e}_{u,v})$$

**Hash Embeddings:** Here, row $m$ corresponds to any value that is assigned to bucket $m$ (i.e. for which $\mathbb{1}_{u,m}$ is one). We find the gradient for the embedding table corresponding to the first feature $V_1$.

$$\nabla_{\mathbf{E}_m} \mathcal{L}_D(\mathbf{E}, \theta) = \sum_{u \in V_1} \mathbb{1}_{u,m} \theta_1 \sum_{v \in V_2} C_{u,v,0} - (C_{u,v,0} + C_{u,v,1}) \sigma_\theta(\mathbf{e}_{u,v})$$

The interpretation of this equation is that with hash collisions, the gradient now contains contributions from any value assigned to the bucket. If we consider the bucket selected by a specfic value $u \in V_1$, we can write this gradient as follows.

$$\nabla_{\mathbf{E}_{h(u)}} \mathcal{L}_D(\mathbf{E}, \theta) = \theta_1 \sum_{v \in V_2} C_{u,v,0} - (C_{u,v,0} + C_{u,v,1}) \sigma_\theta(\mathbf{e}_{u,v})$$
$$+ \theta_1 \sum_{\substack{w \in V_1 \\ w \neq u}} \mathbb{1}_{u,w} \sum_{v \in V_2} C_{u,w,0} - (C_{u,w,0} + C_{u,w,1}) \sigma_\theta(\mathbf{e}_{u,v})$$

**Unified Embeddings:** Here, row $m$ might correspond to any value – even ones from other features. The gradient with respect to $\mathbf{E}_m$ now contains additional components.

$$\nabla_{\mathbf{E}_m} \mathcal{L}_D(\mathbf{E}, \theta) = \sum_{u \in V_1} \mathbb{1}_{u,m} \theta_1 \sum_{v \in V_2} C_{u,v,0} - (C_{u,v,0} + C_{u,v,1}) \sigma_\theta(\mathbf{e}_{u,v})$$
$$+ \sum_{v \in V_2} \mathbb{1}_{v,m} \theta_2 \sum_{u \in V_1} C_{u,v,0} - (C_{u,v,0} + C_{u,v,1}) \sigma_\theta(\mathbf{e}_{u,v})$$

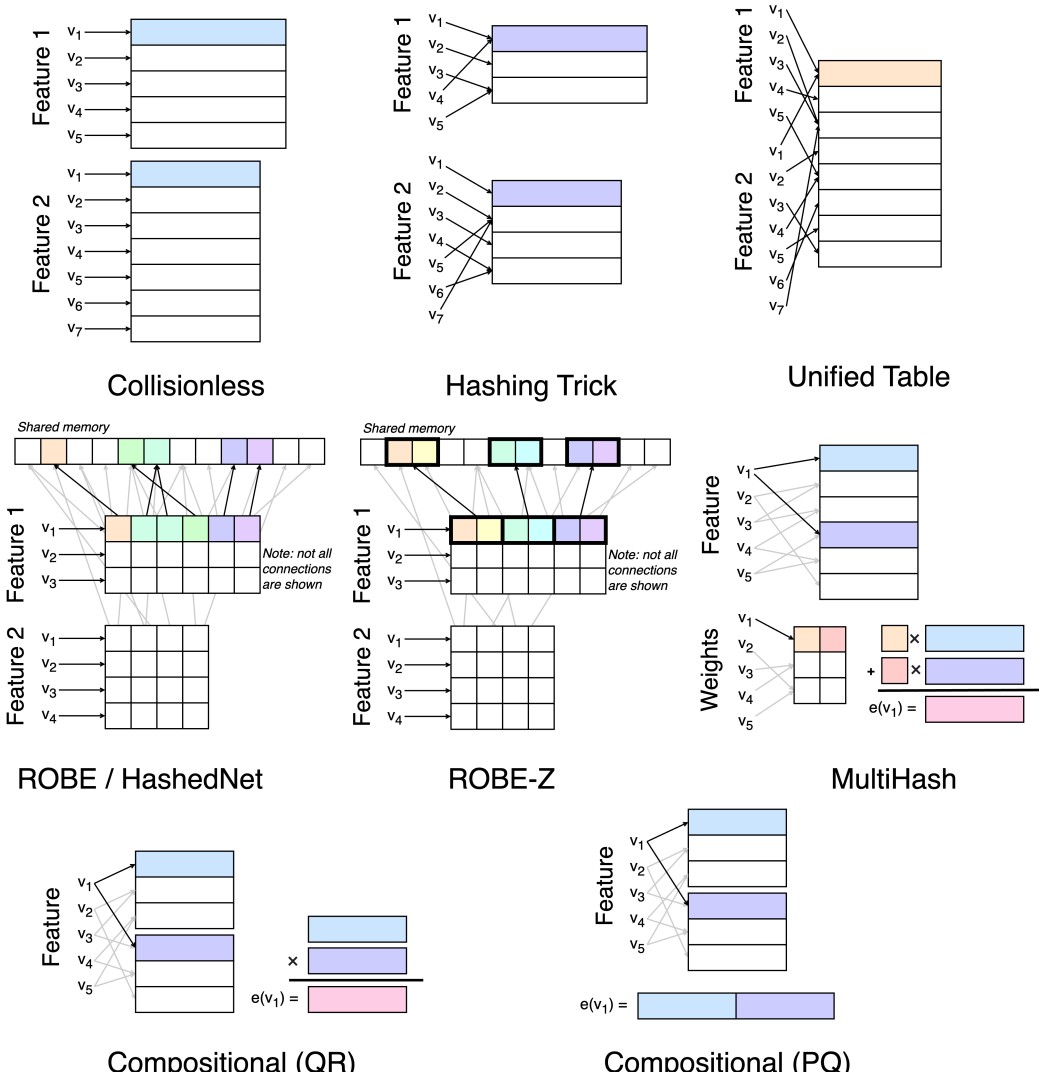

Figure 5: Illustration of baseline embedding methods, all which are compatible with feature multiplexing.

Rewriting the gradient to be in terms of $\mathbf{E}_{h(u)}$ (for $u \in V_1$) reveals that in the unified embedding case, the gradient contains the collisionless component, an intra-feature component, and an inter-feature component:

$$\nabla_{\mathbf{E}_{h(u)}} \mathcal{L}_D(\mathbf{E}, \theta) = \theta_1 \sum_{v \in V_2} C_{u,v,0} - (C_{u,v,0} + C_{u,v,1})\sigma_\theta(\mathbf{e}_{u,v})$$

$$+ \theta_1 \sum_{\substack{w \in V_1 \\ w \neq u}} \mathbb{1}_{u,w} \sum_{v \in V_2} C_{w,v,0} - (C_{w,v,0} + C_{w,v,1})\sigma_\theta(\mathbf{e}_{u,v})$$

$$+ \theta_2 \sum_{v \in V_2} \mathbb{1}_{u,v} \sum_{w \in V_1} C_{w,v,0} - (C_{w,v,0} + C_{w,v,1})\sigma_\theta(\mathbf{e}_{w,u}).$$

## B   Experiments

Figure 5 illustrates our baselines. We describe the methods and their hyperparameters below.

1. Collisionless tables assign each value of each feature a unique $d$-dimensional representation. The only hyperparameter is $d$, the embedding dimension.

2. Hashing trick tables begin with a fixed number of representations and allow values to collide within features. Here, each table has two hyperparameters: $d$ (the dimension) and $M$ (the table size). The inspiration for this method is most commonly attributed to Weinberger et al. (2009), and it is widely used across industry.

3. Unified embeddings use a similar process but allow values to collide within and across features. Unified embeddings have the same hyperparameters as hash tables, but there is now only one table size to tune.

4. Compositional QR tables use multiple small tables for each feature, then perform element-wise multiplication of the resulting embeddings. The hyperparameters are $k$ (the number of tables), $M$ (the table size), and $d$ (the embedding dimension). This method corresponds to the compositional embedding of Shi et al. (2020), using the "concatenation" aggregation strategy.

5. Compositional PQ tables use multiple small tables for each feature, then perform concatenation. We call this "product quantization" because it is similar to the vector compression technique. The hyperparameters are the same as for QR tables. This method corresponds to the compositional embedding of Shi et al. (2020), using the "element-wise product" aggregation strategy.

6. HashedNet tables look up each dimension of each embedding separately in a flat, shared memory. The only extra hyperparameter is the size of the shared parameter array. This is the method of Chen et al. (2015), but with the HashedNet algorithm applied only to the embedding tables.

7. ROBE-Z tables look up contiguous blocks of $Z$ dimensions in shared memory (we use $Z = 2$ in Figure 5). ROBE-Z has two extra hyperparameters: the size of the shared parameter array, and the number of blocks. Note that one can use a different $Z$ for each block. For simplicity, we do not do this. This is the method used by Desai et al. (2022) and by Desai and Shrivastava (2022).

8. Hash embedding tables (or *multihash*, in Figure 5) perform multiple lookups into the same table, then combine the embeddings using a learned weighted combination. Multihash tables have several hyperparameters and are therefore trickier to tune well. The parameters are $M_1$, the size of the embedding table, $M_2$, the size of the importance weight table, $k$, the number of lookups and the dimension of the importance weight table, and $d$, the embedding dimension. This method was proposed by Tsang and Ahle (2022).

**Hyperparameter tuning:**  For all methods, we consider sixteen logarithmically-spaced memory budgets: [0.001, 0.002, 0.005, 0.007, 0.01, 0.02, 0.05, 0.07, 0.1, 0.2, 0.5, 0.7, 1.0, 2.0, 5.0, 10.0] times the memory required for the collisionless table. We conduct 5 independent training runs for each hyperparameter configuration. For non-multiplexed representations, we allocate parameters to features based on vocabulary size. For example, a feature with 10% of the total vocabulary will be allocated 10% of the total parameter budget. We also tuned the network architecture and optimizer parameters. To avoid biasing our results in favor of any one method, we only used collisionless tables to tune these parameters. The embedding hyperparameters were tuned using grid search over the following choices.

- **Hash embeddings**: For hash embeddings (and multiplexed hash embeddings), we chose the number of lookups $k$ from [2, 3]. We also introduce an allocation parameter $p$ that describes what fraction of the memory budget is used for the embedding table (e.g., $M_1$) versus the importance table (e.g., $M_2$). For example, $p = 0.2$ means that the embedding table are allocated 80% of the budget and the importance weight tables receives the remaining 20%. We chose $p$ from [0.05, 0.1, 0.2].

- **ROBE-Z**: We chose the number of lookups from [2, 4, 8, 16]. This corresponds to different values of $Z$ depending on the embedding dimension of the dataset.

- **PQ embeddings**: For PQ embeddings (and multiplexed PQ), we selected the number of lookups $k$ from [2, 3, 4, 8, 16]. We note that this parameter affects performance in different ways depending on the parameter budget – smaller budgets require a greater number of lookups for performance.

- **QR embeddings**: For QR embeddings (and multiplexed QR), we selected the number of components $k$ from [2, 3, 4].

- **Other methods**: Our other methods did not require search over extra hyperparameters. For the special case of collisionless embeddings, we simply ran 5 replicates at a single parameter budget (1.0).

Taken together, we conducted 3205 training runs for each dataset to produce Figure 3. Note that in some cases (e.g., Avazu with a budget of $5, 10$), some runs with expensive configurations did not finish—we report only those results that completed training within 2 days. Finally, the total number of training runs (across all datasets) is somewhat higher, to account for our preliminary tuning over the architecture and optimizer parameters. We estimate that roughly 10,000 training runs were performed in total.

**Criteo:** We use the 7-day version of Criteo from the Kaggle competition. It is widely-known that additional filtering / merging of the vocabulary is necessary to achieve SOTA results on Criteo. We merge the least-frequent values, trimming the vocabulary down to the size listed in Table 4. For the network, we used a stack of two DCN cross layers followed by a 2-layer neural network with 748 nodes in each layer. This model gives a realistic picture of the tradeoff because near-SOTA performance on Criteo can be achieved with a well-tuned 3-layer feedforward network with 748 nodes and collisionless embeddings (Wang et al., 2021). We trained with a batch size of 512 for 300K steps using Adam with a learning rate of 0.0002. The full parameter-accuracy tradeoffs are presented in Figure 7.

**Avazu:** We use the version of Avazu from the Kaggle competition, but we use a different train-test split than the original. Following the practice of Song et al. (2019), we combine the train and test sets and do a 90-10 split after shuffling. We also apply vocabulary pruning to this dataset, with the results in Table 5. Note that we mod the "hour" feature by 24 and do not use the "id" feature. For the neural network, we use a stack of one DCN cross layer followed by two fully connected feedforward layers of 512 nodes each. We trained with a batch size of 512 for 300K steps using Adam with a learning rate of 0.0002. The full parameter-accuracy tradeoffs are presented in Figure 8.

**Movielens:** We use the Movielens-1M dataset with the same preprocessing used by Wang et al. (2021). We only use the "userId," "movieId," "zipcode," "age," "occupation" and "gender" features. We always use a collisionless table to embed the "gender" feature due to small vocabulary size. We used a network composed of a single DCN cross layer followed by a single fully-connected layer with 192 nodes. We trained with a batch size of 128 for 300K steps using Adam with a learning rate of 0.0002. The full parameter-accuracy tradeoffs are presented in Figure 9.

**Vocabulary distributions:** Figure 6 presents the vocabulary distribution for each of our datasets. Criteo differs from Avazu and Movielens in terms of vocabulary distribution - it has a heavier-tailed vocabulary than Avazu or Movielens. This worsens the errors introduced by hash collisions because it is more likely for a colliding token to overwrite the shared embedding (all collisions are between heavy hitters). This is likely the cause of the performance gap between collisionless embeddings (where hash collisions do not occur) and the other methods on Criteo (Figure 3).

**Statistical significance tests:** We conducted additional statistical significance testing on Table 1. When we compare the multiplexed methods against the corresponding non-multiplexed baseline (rather than the strongest baseline), the following differences are significant with $p < 0.05$ (Welch's $t$-test).

1. **Criteo:** All results in the 2.5 MB column. Multiplex Hashing Trick and Multiplex PQ in the 12.5 MB column. Multiplex Hashing Trick, Multiplex HashEmb, Multiplex PQ, and Multiplex QR in the 25MB column.

2. **Avazu:** All results in the 324 kB and 3.24 MB columns. Multiplex QR in the 32.4 MB column.

3. **Movielens:** All results in the 158 kB column. All results in the 791 kB column, except for Multiplex ROBE-Z. Multiplex HashEmb, Multiplex PQ, and Multiplex QR in the 1.6 MB column.

**Hardware and training time:** We used CPU training for our academic experiments. End-to-end CPU training on a shared cluster took approximately 3-4 hours for Criteo, 4-5 hours for Avazu, and 30 minutes for Movielens. The choice of embedding method did not significantly affect the training time.

Table 3: Criteo training time (steps / second).

| Method | Standard | Multiplexed |
|---|---|---|
| Collisionless | 32.9 | NA |
| Hashing Trick | 29.2 | 29.4 |
| (Multiple) Hash Embedding | 33.5 | 31.3 |
| HashedNet | 44.1 | 39.8 |
| ROBE-Z Embedding | 30.0 | 34.4 |
| PQ Embedding | 37.6 | 40.3 |
| QR Embedding | 37.4 | 31.3 |

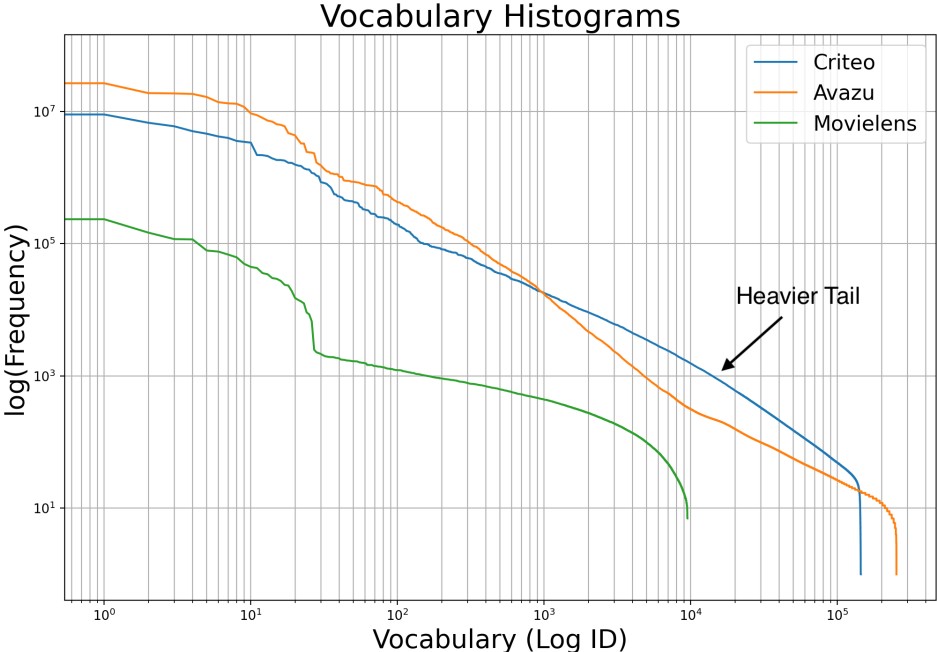

Figure 6: Vocabulary distribution for all categorical feature values (merged) in Criteo, Avazu, and Movielens. All of these datasets display power law behavior, but Criteo has a much heavier tail.

In the Table 3, we report the average number of steps / second for several methods on Criteo for the 25 MB table size with CPU training (higher is better). This is representative of the total training time because the number of steps to convergence is fairly stable across methods (250k batches of size 512 for Criteo, 200K of size 512 for Avazu, and 50K of size 128 for Movielens). Note that because our shared cluster has large variations in job load / demand, these numbers have high variance ($\sigma > 5$). However, our results still support the conclusion that multiplexing has a minimal effect on training time. We observed similar behavior in large-scale industrial experiments—multiplexing did not significantly increase the training or inference time.

### B.1  Industry experiments

**Hardware and training time:**  In several of our Unified Embedding deployments (Table 2), we use TPUv4 for training and/or inference. For models of practical interest, embedding table size rarely affects the model training time in models of practical interest. As long as there is enough memory to support the embedding tables (i.e., enough CPU RAM or total accelerator memory), the training time is mostly governed by the forwards and backwards passes on the rest of the (upstream) network. We do not observe a substantial increase in training / inference time on either CPU or TPUv4 (Jouppi et al., 2023).

**Modeling tradeoffs:**  Unified embedding gives up some flexibility in setting the per-feature embedding dimension, and other (more complicated) multiplexed methods are better on the Pareto-frontier for offline experiments. In exchange, we get more per-feature embedding capacity, a much simpler

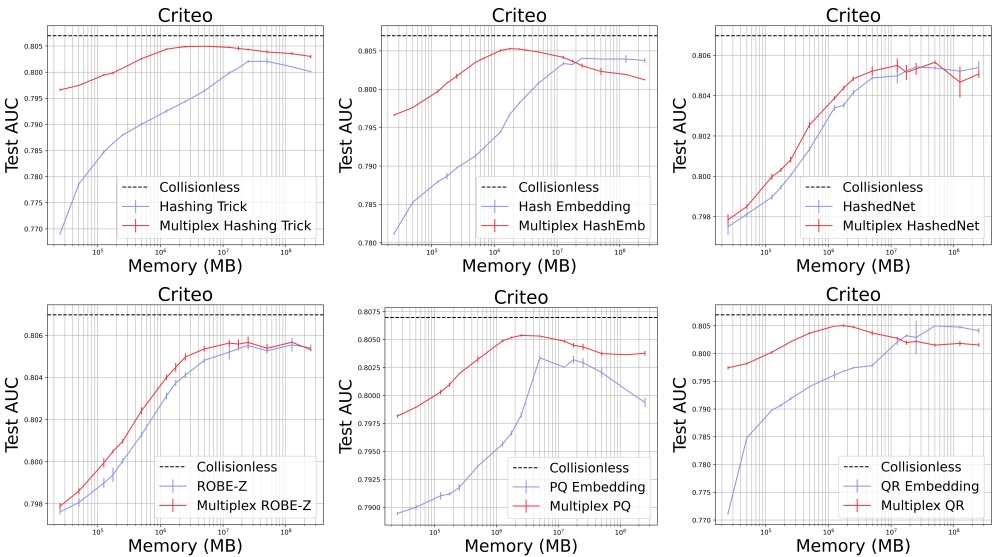

Figure 7: Full parameter-accuracy tradeoffs for all methods on the Criteo dataset. Naming convention is the same as in Table 1. Note the log scale and that plots do not share the y-axis scale. The error bar width denotes the standard deviation over 5 model training runs.

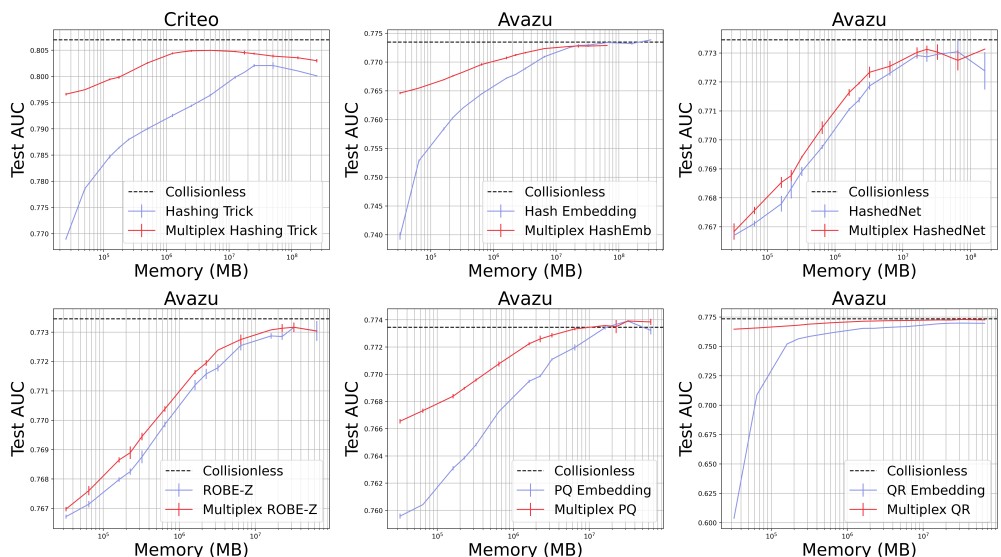

Figure 8: Full parameter-accuracy tradeoffs for all methods on the Avazu dataset. Naming convention is the same as in Table 1. Note the log scale and that plots do not share the y-axis scale. The error bar width denotes the standard deviation over 5 model training runs.

hyperparameter configuration, hardware-friendly algorithm, and easy feature engineering. For example, to add new features to a model without multiplexing, we must specify and tune the table size and dimension for a new set of embedding tables. With multiplexing, we can simply add a new feature to an existing table at the cost of a few additional lookups. Another option is to use the conventional scheme for new features, then regularly consider a model update that migrates several new features into the unified embedding (typically resulting in performance gains).

**Engineering tradeoffs:** During our experience deploying Unified Embedding in a dozen applications, we have not had problems with observability, monitoring, or model health / stability. Instead, we find that Unified Embedding results in better load balancing and alleviate TPU hot-spot issues (as our

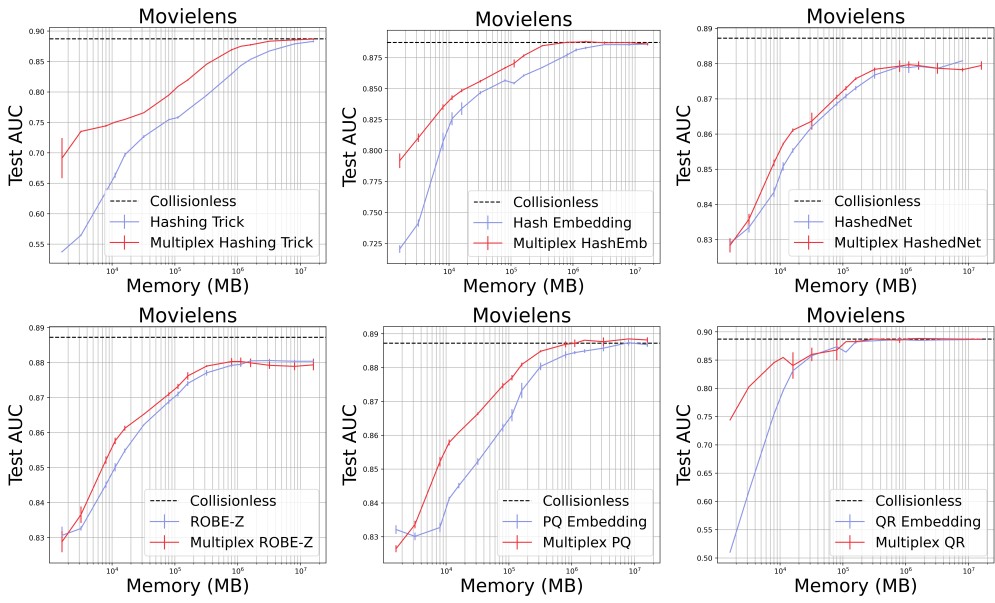

Figure 9: Full parameter-accuracy tradeoffs for all methods on the Movielens-1M dataset. Naming convention is the same as in Table 1. Note the log scale and that plots do not share the y-axis scale. The error bar width denotes the standard deviation over 5 model training runs.

embedding rows are distributed over cores). The high-level takeaway is that it is easier to balance one large table across accelerators than many small tables of different shapes.

**Model health:** All of our industry online experiments assume healthy models as a prerequisite. These models use online sequential training and adapt to constant variations in data distributions. The consistent improvements from unified embedding in these systems is strong evidence that our methods do not interfere with model health or add maintenance costs.

Table 4: Criteo vocabulary pruning.

| Feature | Vocabulary Size |
|---|---|
| 14 | 676 |
| 15 | 533 |
| 16 | 17447 |
| 17 | 19995 |
| 18 | 180 |
| 19 | 13 |
| 20 | 9693 |
| 21 | 337 |
| 22 | 3 |
| 23 | 14637 |
| 24 | 4378 |
| 25 | 17795 |
| 26 | 3067 |
| 27 | 26 |
| 28 | 6504 |
| 29 | 18679 |
| 30 | 10 |
| 31 | 3102 |
| 32 | 1557 |
| 33 | 3 |
| 34 | 18230 |
| 35 | 10 |
| 36 | 14 |
| 37 | 13079 |
| 38 | 56 |
| 39 | 10581 |

Table 5: Avazu vocabulary pruning.

| Feature | Vocabulary Size |
|---|---|
| C1 | 8 |
| C14 | 2309 |
| C15 | 9 |
| C16 | 10 |
| C17 | 405 |
| C18 | 5 |
| C19 | 66 |
| C20 | 167 |
| C21 | 56 |
| app_category | 29 |
| app_domain | 277 |
| app_id | 4438 |
| banner_pos | 8 |
| device_conn_type | 5 |
| device_id | 67767 |
| device_ip | 163804 |
| device_model | 6217 |
| device_type | 6 |
| site_category | 24 |
| site_domain | 3887 |
| site_id | 3317 |
| hour | 24 |

