# OpenReview forum: "Unified Embedding: Battle-Tested Feature Representations for Web-Scale ML Systems"
_NeurIPS.cc/2023/Conference — NeurIPS 2023 spotlight_

### Official Review · Reviewer_5UVX · 2023-06-17

**Soundness:** 3 good
**Presentation:** 3 good
**Contribution:** 3 good
**Rating:** 7
**Confidence:** 4

**Summary:**

This paper introduces a novel approach known as Feature Multiplexing, which allows multiple features to share a single representation space in machine learning systems. This is significant for web-scale systems which handle hundreds of features with vocabularies of up to billions of tokens, where the standard embedding approaches introduce a vast number of parameters. The authors propose a new solution called Unified Embedding, which simplifies feature configuration, adapts to dynamic data distributions, and is compatible with modern hardware. The empirical results from multiple web-scale search, ads, and recommender systems show superior performance compared to highly competitive baselines.

**Strengths:**

1) The paper addresses a crucial problem in large-scale machine learning systems related to efficient and effective learning of feature embeddings. The proposed framework, Feature Multiplexing, is innovative, allowing multiple features to share the same representation space.

2) The authors provide a well-written and clear explanation of the concepts and the proposed solution. The paper is well-structured, with a good balance of theory, experimentation, and discussion.

3) The problem this paper addresses is of substantial significance, considering the scale at which modern machine learning systems operate. The introduction of Unified Embedding could lead to substantial advancements in web-scale ML systems, serving billions of users globally.

**Weaknesses:**

1) The paper lacks details regarding the computational benefits of the proposed technique, specifically in terms of infrastructure gains, parameter size, hardware usage, and training time. Providing such details would make the comparison to the baseline more comprehensive and persuasive. (particularly the large scale experiments explained at the end)

2) Some specific analysis and explanations are missing. For example, why the Criteo dataset behaves differently from Avazu and Movielens is not explained. A more in-depth exploration would strengthen the understanding of the behavior of the proposed technique across datasets.

3) The authors could have provided more insights into why online deployment results are providing gains. A detailed explanation could better support the claim of real-world applicability and insights into future users.

**Questions:**

1) Could the authors provide more information on the impact of Feature Multiplexing and Unified Embedding on the ML infrastructure, particularly in terms of computational costs, training time, and hardware usage?

2) Can the authors elaborate on the specific behaviors of the Criteo dataset compared to Avazu and Movielens datasets?

3) Could the authors discuss the limitations and potential trade-offs of the proposed method? How might it affect the ease of extending the model and conducting future R&D?

**Limitations:**

The authors have not adequately addressed the limitations and potential trade-offs of their proposed technique. Future work may be constrained or impacted by these unaddressed issues. For instance, the authors have not discussed the ease (or lack thereof) of extending the model with new features or conducting new R&D with the proposed method. They also have not explored the potential maintenance costs and impacts on model health and observability, which can be crucial for deploying such systems in real-world applications. Further information on these aspects could greatly benefit the audience's understanding of the practical viability and potential challenges of implementing the proposed technique.

---

> ### Author Rebuttal · Authors · 2023-08-09
>
> Thank you for your review. We address the weaknesses and questions below.
>
> **W1 and Q1:**
> - **Regarding training time:** Embedding table size rarely affects the model training time in models of practical interest. As long as there is enough memory to support the embedding tables (i.e., enough CPU RAM or total accelerator memory), the training time is mostly governed by the forwards and backwards passes on the rest of the (upstream) network.
>
> - In the following table, we report the average number of steps / second for several methods on Criteo for the 25 MB table size with CPU training (higher is better). This is representative of total training because the number of steps to convergence is fairly stable across methods (250k batches for Criteo, 200K for Avazu, and 50K for Movielens).
> |Method|Standard|Multiplexed|
> |--|--|--|
> |Collisionless|32.9|NA|
> |Hashing Trick|29.2|29.4|
> |(Multiple) Hash Embedding|33.5|31.3|
> |HashedNet|44.1|39.8|
> |ROBE-Z Embedding|30.0|34.4|
> |PQ Embedding|37.6|40.3|
> |QR Embedding|37.4|31.3|
>
> - Note that because our shared cluster has large variations in job load / demand, these numbers have high variance ($\sigma > 5$). However, these results still support the conclusion that multiplexing has a minimal effect on training time. We observed similar behavior in large scale experiments (see caption of Table 2).
>
> - **Hardware:** We use proprietary ML accelerators with dedicated embedding support - see [TPUv4](https://arxiv.org/pdf/2304.01433.pdf) and [MITA v1](https://ai.meta.com/blog/meta-training-inference-accelerator-AI-MTIA/) for examples of this kind of system (to preserve anonymity during review, we will disclose the full details after the review process). For our academic evaluations, end-to-end CPU training took approximately 3-4 hours for Criteo, 4-5 hours for Avazu, and 30 minutes for Movielens.
>
> **W2 and Q2:** This is a good question. Criteo differs from Avazu and Movielens in terms of vocabulary distribution - it has a heavier-tailed vocabulary than Avazu or Movielens. This worsens the errors introduced by hash collisions because it is more likely for a colliding token to overwrite the shared embedding (all collisions are between heavy hitters). This is likely the cause of the performance gap between collisionless embeddings (where hash collisions do not occur) and the other methods on Criteo. See the rebuttal PDF for a plot of the vocabulary distributions.
>
> **W3:** In short, online systems in industry are regularly memory-bound. We can almost always improve model quality with larger embedding tables, but this requires more CPU/GPU/accelerator memory and is not worth the resource cost/revenue trade-off. Improving model quality for the same memory budget is an immediate win.
> Our offline experiments (from Table 1 / Figure 3) show that multiplexed embeddings lift the Pareto frontier, allowing us to either have better performance at a fixed size or equal performance with a smaller size. The following evidence suggests that this tradeoff drives our online performance gains:
> - Increases in model capacity often lead to better online performance (see [“Scaling Law for Recommendation Models”](https://arxiv.org/pdf/2111.11294.pdf)). Our theory and offline experiments show that multiplexing increases the effective capacity of a model, allowing a smaller model to behave like a larger one.
> - We observe (line 353- 355) that multiplexing provides the greatest benefits for problems with large and/or dynamic vocabulary. These are exactly the applications known to require higher model capacity (see [“Learning to Embed Categorical Features without Embedding Tables for Recommendation”](https://arxiv.org/abs/2010.10784)).
>
> **Q3:** While we discuss some limitations in the text (inline, due to the page limit), we agree that a description of our engineering and modeling tradeoffs would be beneficial to include. In the revision, we plan to summarize these points in a clearly-marked limitations section and provide a full discussion in the appendix. Thank you for raising these valuable questions.
>
> **Modeling tradeoffs:** Unified embedding gives up some flexibility in setting the per-feature embedding dimension, and other (more complicated) multiplexed methods are better on the Pareto-frontier for offline experiments. In exchange, we get more per-feature embedding capacity, a much simpler hyperparameter configuration, hardware-friendly algorithm, and easy feature engineering. For example, to add new features to a model without multiplexing, we must specify and tune the table size and dimension for a new set of embedding tables. With multiplexing, we can simply add a new feature to an existing table at the cost of a few additional lookups. Another option is to use the conventional scheme for new features, then regularly consider a model update that migrates several new features into the unified embedding (typically resulting in performance gains).
>
> **Engineering tradeoffs:** During our experience deploying unified embedding in a dozen applications, we have not had problems with observability, monitoring, or model health / stability. Instead, we find that unified embeddings have better load balancing and alleviate hot-spot issues (our embedding rows are distributed over cores). The high-level takeaway is that it is easier to balance one large table across accelerators than many small tables of different shapes.
>
> **Model health:** All of the industry online experiments in Table 2 assume healthy models as a prerequisite. These models train online and adapt to constant variations in data distributions. The consistent improvements from unified embedding in these systems is strong evidence that our methods do not interfere with model health or add maintenance costs.
>
> Finally, thank you for taking the time to do a detailed review of our paper. If you feel that we have addressed your concerns, we hope that you will consider raising the score.

---

> > ### Comment · Reviewer_5UVX · 2023-08-18
> >
> > Thanks to the authors for detailed response. I think crucial and adding a lot of practical value to the paper and impact. I strongly recommend these comments to be embeded into the final version. I am happy to raise my score as well. I recommend accept.

---

### Official Review · Reviewer_fp86 · 2023-07-09

**Soundness:** 3 good
**Presentation:** 3 good
**Contribution:** 3 good
**Rating:** 7
**Confidence:** 3

**Summary:**

The authors present a method for multiplexing embeddings of various features in the recommender and similar applications, i.e., sharing the feature embeddings in order to save space and improve performance, especially at lower memory budgets. They provide a detailed overview of the relevant prior work, and give strong both theoretical and empirical analysis of the proposed method. They show the benefits of the method on three public data sets, and also show how the method helped in large-scale production setting.

**Strengths:**

- Very important problem being addressed.
- Good theoretical discussion of the method.
- Good results shown in the production-level setup.

**Weaknesses:**

- In some places the explanations can be improved quite a lot.
- The results are very mixed in some cases.

**Questions:**

Please find the detailed comments below:
- The data set details should be explained better. E.g., in Fig 2 the authors mention 26 Criteo features, and it is unclear which ones they are referring to. This is adding to some confusion.
- This holds also for other data sets. that are considered. Adding more details there would make the experimental section much more readable.
- The results in Table 1 show that the method is only outperforming other baselines on low-budget Criteo, while everywhere else it is worse. This is quite a weak result, and it is unclear why would someone use their method with such mixed performance.
- Multiplexed versions of the other baselines should be better explained, at least briefly. The methods are just given, without much discussion.

*************** UPDATE AFTER REBUTTAL **************
I would like to thank the authors for their responses. They do address my concerns, and I am happy to increase the score.

**Limitations:**

The authors did not discuss the limitations, and it would be good to add a short paragraph.

---

> ### Author Rebuttal · Authors · 2023-08-09
>
> Thank you for your review. We address the questions and weaknesses below. In particular, we think there is a misunderstanding about our experiment results (which we would like to clarify).
>
> **Regarding datasets:** This is a good point - we agree that it would be valuable to have more detailed explanations here. We will elaborate in the revision. To clarify the confusion about Figure 2, Criteo is an online advertisement dataset of about 45 million examples (7 days of data), where each example corresponds to a user interaction (click/no-click decision) and contains an additional 39 features. 26 of these features are categorical and 13 of them are real-valued continuous variables. Figure 2 refers to the 26 categorical features in Criteo (the continuous variables are typically fed alongside the embedded categorical features, but do not use an embedding table - see the [DLRM paper](https://arxiv.org/abs/1906.00091) and the related literature for details).
>
> For the other datasets: Avazu is a collection of 11 days of advertisement click data (approximately 36 million examples), where each example contains 23 categorical features and no continuous features. Movielens is a traditional user-item recommendation problem, very similar to the well-known Netflix Prize task. These are all highly popular datasets that are widely used to evaluate recommendation algorithms. We plan to refer the reader to "BarsCTR: Open Benchmarking for Click-Through Rate Prediction," ([CIKM 2021](https://arxiv.org/abs/2009.05794)) for a very detailed description of Avazu and Criteo and to "The Movielens Datasets: History and Context" (ACM Transactions on Interactive Intelligent Systems, 2015) for a thorough description of Movielens.
>
> **Regarding experiments:** We think there is a misunderstanding here. Table 1 shows that multiplexed embeddings (our proposal) outperform all of the baselines at all memory budgets on all datasets (except for collisionless embeddings, which are infeasible in practice and included only as a headroom reference point - see the response to Reviewer Azh8 for a detailed discussion of this matter). Note that all of the methods below the horizontal bar are newly proposed, while existing baselines are listed above the bar. Figure 3 shows the top baseline (blue) against the top multiplexed method (red) and is another view of the same data behind Table 1.
>
> In practice, we use Multisize Unified Embeddings (see line 306), which are very similar to Multiplex PQ - one of the top performers from Table 1. We would also like to highlight that in large-scale industrial applications, the extreme low-budget case is standard practice because vocabulary sizes regularly exceed 10 billion, making collisionless schemes infeasible. Our performance improvement in this memory-constrained setting is therefore a major advantage. To summarize, all of the methods below the bar in Table 1 are ours (and are new), and we show very strong results in real-world settings where the vocabulary is on the order of millions to billions (Table 2).
>
> To remove this confusion, we plan to revise the names to be clearer and more descriptive:
> - What was previously called “Multisize Unified Embeddings” (in Section 5.2) will be referred to simply as “Unified Embedding.”
> - What was previously called “Unified Embedding” (in Table 1) will be referred to as “Multiplexed Hashing Trick.”
>
> See the rebuttal PDF for a draft revision of Table 1.
>
> **Regarding other multiplexed methods:** We describe the last 20 years of SOTA embedding methods in the appendix, but we agree that it would be valuable to have a detailed description of how to construct a multiplexed method given an existing embedding strategy. We will add the following description:
>
> “To construct a multiplexed version of an existing embedding scheme, we use a single instance of the scheme to index all of the features. For example, we can look up all feature values in a single multihash table, rather than using a separately-tuned multihash table for each feature. Practically, this is equivalent to salting the vocabulary of each feature with the feature ID and merging the salted vocabularies into a single, massive categorical vocabulary (which is then embedded as usual).”
>
> **Regarding limitations:** Due to space constraints, we discussed limitations throughout the text. Limitations of the theory are addressed in Section 4.3, modeling limitations (due to embedding width constraints, etc) in Section 3 (around line 150), and the constraints of our experimental setup in Section 5.
>
> However, after reading the reviews, it seems that this was very easy to miss. We plan to add the following paragraph to the revision:
>
> *Limitations:* While we expect deeper and more complicated network architectures to exhibit similar behavior, our theoretical analysis is limited to single-layer neural networks. The Pareto frontier from Section 5.1 is based on models that lag behind the current SOTA (we use 1-2 DCN layers + 1-2 DNN layers), though we do provide evidence that feature multiplexing is also effective for SOTA models in Table 2. Unified embeddings impose limitations on the overall model architecture, and all embedding dimensions must share a common multiple. Because of the latency involved with looking up more than 6 components, we are effectively limited to 5-6 discrete choices of embedding width. Finally, unified embeddings sacrifice the ability to aggressively tune hyperparameters on a per-feature basis in exchange for flexibility, ease-of-use, and simplified feature engineering.
>
> Finally, thank you for taking the time to review our paper. If you feel that we have addressed your concerns, we hope that you will consider raising the score.

---

> > ### Comment · Reviewer_fp86 · 2023-08-21
> >
> > I would like to thank the authors for their responses. They do address my concerns, and I have increased my score.

---

### Official Review · Reviewer_Azh8 · 2023-07-27

**Soundness:** 3 good
**Presentation:** 4 excellent
**Contribution:** 3 good
**Rating:** 6
**Confidence:** 3

**Summary:**

This paper proposes that in web-scale machine learning systems, features from different fileds can share the embedding matrix without significantly affecting the model's performance. The insight lies in that different feature fields are processed by different model parameters. Therefore, compared to inter-feature collisions, in the case of intra-feature collisions, the embeddings of different features can tend to be orthogonal, which is beneficial to the final model performance. This point is confirmed by the empirical study based on logistic regression. The experimental results show that the multiplexed embedding scheme is more effective than the existing schemes that only consider inter-feature embedding sharing.

------
AFTER REBUTTAL:

I've read and appreciated the author’s rebuttal. I understand that collisionless embeddings are considered the upper bound. However, it would be helpful if the advantages of feature multiplexing could be mentioned and demonstrated in the discussion and experiments. Thus I would like to keep the score.

**Strengths:**

- The overall organization and writing of the paper are excellent.
- The proposed multiplexed feature embedding scheme is novel and its feasibility is verified both theoretically and experimentally.
- The paper conducts experiments on multiple public datasets, and the overall results are promising.

**Weaknesses:**

- In the experiments, although the performance is better compared to the inter-feature embedding sharing scheme, it is worse than the Collisionless scheme on Criteo, very close on Avazu, but no relevant discussion is provided.
- Compared to the Collisionless scheme, the advantages of using feature multiplexing do not seem to be fully discussed, nor are they reflected in the experimental results.

**Questions:**

Please share your views on the aforementioned weaknesses.

**Limitations:**

Yes.

---

> ### Author Rebuttal · Authors · 2023-08-09
>
> Thank you for your review. We address the questions raised in review below.
>
> **Regarding experiment results:** There seems to be a misunderstanding about collisionless embeddings. This method is not usually a feasible baseline, and it is included in our benchmark evaluation only as a headroom reference point. In industrial recommendation systems, where feature vocabularies can be massive and change dynamically due to churn, the hashing trick is the standard approach (e.g. Google [1], Facebook [2], Twitter [3]). Because of memory limits and dynamic vocabularies that are not fully known up-front, collisionless tables are often impossible to deploy.
>
> For example, our pCTR application in Table 2 would require > 1 TB of memory for a 32-dimensional collisionless table. Facebook’s DLRM model [4] requires a 40 TB table even after hashing. While Facebook does not disclose their total vocabulary size, a collisionless table would likely require space that is 1-2 orders of magnitude larger (> 100 TB in their case). We conducted our academic experiments on the 7-day Criteo dataset, where the collisionless table only requires 25 MB. When more realistic time periods are considered, the size of the vocabulary grows by several orders of magnitude (e.g., the collisionless table for the 30-day Criteo dataset is larger than 100 GB [5]).
>
> Hence, the hashing-based methods are the only realistic baselines in our evaluation. Collisionless embedding performance effectively represents an upper bound on what is practically attainable. We include them in our (small-scale) experiments as an optimistic reference point, to investigate the Pareto frontier and estimate the ideal model capacity when we are not limited by hash collisions. The experiments show that multiplexing achieves close to full capacity on Movielens / Avazu and provides the best performance on Criteo versus all baselines.
>
> **Criteo vs. Avazu / Movielens:** The performance gap on Criteo is likely a result of the vocabulary distribution. Criteo has a heavier-tailed vocabulary than Avazu or Movielens. This worsens the errors introduced by hash collisions because it is more likely for a colliding token to overwrite the shared embedding (all collisions are between heavy hitters). We have included a plot of the vocabulary distribution in the rebuttal PDF, which we hope will shed some light on this point.
>
> **Regarding our advantages:** We cover the algorithm-level advantages of feature multiplexing in Section 3 and the practical advantages in Section 5.2. Compared to collisionless embedding, these approaches are fast, implementable, and orders of magnitude cheaper to train and serve.
>
> We will add this discussion to the revision. Once again, thanks for reviewing our work. If we have addressed your questions and the weaknesses identified in the review, we hope that you will consider raising the score.
>
> **References**
> - [1] Learning to Embed Categorical Features without Embedding Tables for Recommendation (https://arxiv.org/pdf/2010.10784.pdf)
> - [2] Compositional Embeddings Using Complementary Partitions for Memory-Efficient Recommendation Systems (https://arxiv.org/pdf/1909.02107.pdf)
> - [3] "Model Size Reduction Using Frequency Based Double Hashing for Recommender Systems" [RecSys 2020](https://arxiv.org/pdf/2007.14523.pdf)
> - [4] "Software-Hardware Co-design for Fast and Scalable Training of Deep Learning Recommendation Models" [ISCA'22](https://arxiv.org/abs/2104.05158)
> - [5] "The trade-offs of model size in large recommendation models: 100GB to 10MB Criteo-tb DLRM model" [NeurIPS 22](https://arxiv.org/abs/2207.10731)

---

### Official Review · Reviewer_fvbA · 2023-08-02

**Soundness:** 3 good
**Presentation:** 3 good
**Contribution:** 3 good
**Rating:** 5
**Confidence:** 4

**Summary:**

The paper introduces a novel "Feature Multiplexing" framework which uses a shared representation space (embedding table) for multiple sparse features. This approach aims to find a balance between model size and accuracy for industrial level recommender system. Besides, the authors provide a theoretical analysis, highlighting that inter-feature collisions can be alleviated if features are projected using orthogonal weight vectors. Further gradient analysis reveals that these collisions are not uniformly detrimental; the adverse effects can be mitigated when features are processed by distinct parameters in a single-layer neural network.

**Strengths:**

**Pros**:

- The paper presents the a novel "Feature Multiplexing" framework, offering a straightforward and effective method to optimize the trade-off between model size and accuracy.

- This framework promises considerable practical advantages, especially in the context of large-scale recommendation systems.

- A  theoretical analysis is provided, addressing the advantage of the Feature Multiplexing system from rigorous theoretical analysis . It explains how inter-feature collisions can be reduced when the model uses orthogonal weight vectors to project distinct features. This analysis provide good theoretical ground for a fairly practical design.

**Weaknesses:**

**Cons**:

- Certain claims in the paper, such as "0.02% increase in test AUC is considered significant in Avazu and Criteo" and "+0.1% is considered significant in online systems," raise eyebrows. These claims appear to be based on subjective opinions rather than objective facts.

- The authors have not provided source code for their work. This factor makes it difficult to reproduce and verify the claims made in the paper, which is a foundational principle of the NeurIPS community. If this work is industry-driven and the code cannot be released easily, perhaps the authors should consider venues more suited to applied data mining.

- The content and focus of the paper lean heavily towards data mining and address real-world, industrial-scale problems. As such, it might be better suited for venues like KDD, WWW, or SIGIR.


**Questions:**

Please refer to weakness.

**Limitations:**

The authors did not discuss the limitations at all.

---

> ### Author Rebuttal · Authors · 2023-08-09
>
> Thank you for your review. We address the weaknesses below.
>
> **Regarding significance of results:** It is completely fair to be skeptical about the significance of a +0.1% improvement. However, seemingly small improvements can translate to huge numbers in large online systems. For example, Anil et al. state that “accuracy improvements above 0.1% are considered significant” in their [paper](https://arxiv.org/abs/2209.05310) “On the Factory Floor: ML Engineering for Industrial-Scale Ads Recommendation Models'' when discussing Google Ads models. This is an industry with $200B+ annual revenue. In “BarsCTR: Open Benchmarking for Click-Through Rate Prediction” ([CIKM 2021](https://arxiv.org/abs/2009.05794)), Zhu et al. summarize the literature: “existing studies from Google [8, 47] and Microsoft [28] reveal that an absolute improvement of 0.1% in logloss (or AUC) is considered as practically significant in real CTR prediction problems.” All of the results in Table 2 are statistically significant with $p < 0.05$, and 0.1% is very practically significant in industrial systems.
>
> This is also true in academic evaluations: the differences between logistic regression and SOTA on Avazu and Criteo are +1.7% and +2.1% AUC, respectively (see BarsCTR). The [DCNv2 paper](https://arxiv.org/abs/2008.13535) states, “For Criteo, a 0.001-level improvement is considered significant (see [13, 46, 50]).” The [AutoInt paper](https://arxiv.org/abs/1810.11921) claims that “a slightly higher AUC or lower LogLoss at 0.001-level is regarded significant for CTR prediction task.” Our +0.02% AUC number from Table 1 is based on the 95% confidence interval surrounding the mean AUCs, with $\sigma = 0.00022$ (approximate population standard deviation, estimated via ~20 runs). However, we really should have made this much clearer, so thank you for making this point in your review. In the revision, we have performed an unpaired t-test between the values in Table 1, finding most differences to be statistically significant. You can find the revised table in the rebuttal PDF.
>
> **Regarding code:** We are in the process of releasing an open source implementation for the academic experiments to reproduce Table 1. Due to our internal publication review process, it wasn’t possible to release this by the paper submission deadline, but it will be present in the revision.
>
> **Regarding venue:** Embedding learning (or representation learning) is the heart of deep learning, and is a problem that occurs in NLP, vision, and bioinformatics as well as in data mining. Like many recent papers (e.g. ROBE-Z at NeurIPS 2022), we focus on SAR (search, ads, and recommendation) systems because they are the most compelling application of categorical representation learning. However, feature multiplexing is a general technique with many potential applications. Applications of embedding learning include scaling up the vocabulary size in transformers and improving the compute-memorization tradeoff of embedding retrieval-augmented transformers and LLMs. Finally, we note that the top baseline methods for this paper recently appeared at NeurIPS (Multihash Hash Embeddings in 2017, ROBE-Z in 2022) and similar venues (HashedNets and Hashing Trick at ICML).
>
> **Regarding limitations:** Due to space constraints, we discussed limitations throughout the text. Limitations of the theory are addressed in Section 4.3, modeling limitations (due to embedding width constraints, etc) in Section 3 (around line 150), and the constraints of our experimental setup in Section 5.
>
> However, after reading the reviews, it seems that this was easy to miss. We agree that it would be clearer to consolidate this in a separate section. We’re happy to add the following paragraph to the revision:
>
> *Limitations:* While we expect deeper and more complicated network architectures to exhibit similar behavior, our theoretical analysis is limited to single-layer neural networks. The Pareto frontier from Section 5.1 is based on models that lag behind the current SOTA (we use 1-2 DCN layers + 1-2 DNN layers), though we do provide evidence that feature multiplexing is also effective for SOTA models in Table 2. Unified embeddings impose limitations on the overall model architecture, and all embedding dimensions must share a common multiple. Because of the latency involved with looking up more than 6 components, we are effectively limited to 5-6 discrete choices of embedding width. Finally, unified embeddings sacrifice the ability to aggressively tune hyperparameters on a per-feature basis in exchange for flexibility, ease-of-use, and simplified feature engineering.
>
> Once again, thank you for taking the time to review our paper. If we have addressed the questions and weaknesses from your review, we hope that you will consider raising the score.

---

### Author Rebuttal · Authors · 2023-08-09

We'd like to thank all of the reviewers for their efforts. We have included a rebuttal PDF that contains:
- Vocabulary distributions for Critoe, Avazu, and Movielens. This plot helps to explain the differences in behavior for embedding algorithms on Criteo vs Avazu and Movielens.
- A revised version of Table 1 that we hope will clear up some common confusions about our results. We would like to highlight that multiplexed embedding algorithms (our proposal) show improvements across the full memory-accuracy tradeoff for each of our three datasets. The revision of Table 1 also includes results for t-tests at the $p < 0.01$ and $p < 0.05$ significance levels, which demonstrate statistically significant differences between multiplexed vs. non-multiplexed embeddings.

Please see the individual rebuttals for responses to specific reviewer questions.

---

### Decision · Program_Chairs · 2023-09-21

**Decision:**

Accept (spotlight)

**Comment:**

The paper provides an approach to the problem of feature embedding, which is a crucial part in several ML designs, and in particular in search ads and recommendation systems (called SAR in the paper). The core idea, called feature multiplexing, is sensible and effective. The reviews agree that the application is novel, and appreciated the theoretical analysis provided. Furthermore, the experiments are overall agreed to show the effectiveness of the proposed method and its advantage over existing work. The reviews pointed out a need for a better discussion or explanation of the results, but these seem to be focused on particular points and the authors’ response mitigated these concerns. I think that after editing based on the discussions here, the paper can be made ready to be published in NeurIPS. The novelty of the work and wide impact of the technique would make it a welcome addition to the venue.